# Genomic Characteristics and Molecular Epidemiology of Multidrug-Resistant *Klebsiella pneumoniae* Strains Carried by Wild Birds

Xue Wang,[a] Jianan Zhao,[a] Fang Ji,[a] Meng Wang,[a,b] Bin Wu,[a] Jianhua Qin,[b] Guoying Dong,[c] Ruili Zhao,[d] Chengmin Wang[a]

[a]Guangdong Key Laboratory of Animal Conservation and Resource Utilization, Institute of Zoology, Guangdong Academy of Science, Guangzhou, China
[b]College of Veterinary Medicine, Agricultural University of Hebei, Baoding, China
[c]College of Global Change and Earth System Science, Beijing Normal University, Beijing, China
[d]College of Animal Science and Veterinary Medicine, Tianjin Agricultural University, Tianjin, China

Xue Wang and Jianan Zhao contributed equally to this work. Author order was determined both alphabetically and in order of increasing seniority.

**ABSTRACT** This study aimed to explore the relationship between wild birds and the transmission of multidrug-resistant strains. *Klebsiella pneumoniae* was isolated from fresh feces of captured wild birds and assessed by the broth microdilution method and comparative genomics. Four *Klebsiella pneumoniae* isolates showed different resistance phenotypes; S90-2 and S141 were both resistant to ampicillin, cefuroxime, and cefazolin, while M911-1 and S130-1 were sensitive to most of the 14 antibiotics tested. S90-2 belongs to sequence type 629 (ST629), and its genome includes 30 resistance genes, including $bla_{CTX-M-14}$ and $bla_{SHV-11}$, while its plasmid pS90-2.3 (IncR) carries *qacEdelta1*, *sul1*, and *aph(3')-Ib*. S141 belongs to ST1662, and its genome includes a total of 27 resistance genes, including $bla_{SHV-217}$. M911-1 is a new ST, carrying $bla_{SHV-1}$ and *fosA6*, and its plasmid pM911-1.1 (novel) carries *qnrS1*, $bla_{LAP-2}$, and *tet*(A). S130-1 belongs to ST3753, carrying $bla_{SHV-11}$ and *fosA6*, and its plasmid pS130-1 [IncFIB(K)] carries only one resistance gene, *tet*(A). pM911-1.1 and pS90-2.3 do not have conjugative transfer ability, but their resistance gene fragments are derived from multiple homologous *Enterobacteriaceae* strain chromosomes or plasmids, and the formation of resistance gene fragments (multidrug resistance region) involves interactions between multiple mobile element genes, resulting in a complex and diverse resistance plasmid structure. The homologous plasmids related to pM911-1.1 and pS90-2.3 were mainly from isolated human-infecting bacteria in China, namely, *K. pneumoniae* and *Escherichia coli*. The multidrug-resistant *K. pneumoniae* isolates carried by wild birds in this study had drug resistance phenotypes conferred primarily by multidrug resistance plasmids that were closely related to human-infecting bacteria.

**IMPORTANCE** Little is known about the pathogenic microorganisms carried by wild animals. This study found that the multidrug resistance phenotype of *Klebsiella pneumoniae* isolates carried by wild birds was mainly attributed to multidrug resistance plasmids, and these multidrug resistance plasmids from wild birds were closely related to human-infecting bacteria. Wild bird habitats overlap to a great extent with human and livestock habitats, which further increases the potential for horizontal transfer of multidrug-resistant bacteria among humans, animals, and the environment. Therefore, wild birds, as potential transmission hosts of multidrug-resistant bacteria, should be given attention and monitored.

**KEYWORDS** wild birds, multidrug-resistant bacteria, *Klebsiella pneumoniae*, drug resistance plasmids

Address correspondence to Chengmin Wang, wangchm@giz.gd.cn.

The authors declare no conflict of interest.

10.1128/spectrum.02691-22 **1**

*K*lebsiella pneumoniae has long been considered a pathogen and is still one of the most common nosocomial pathogens in the world (1). It is widespread in Asia, Africa, and Europe, causing tens of thousands of infections and deaths every year (2–4). Notably, *K. pneumoniae* can occupy favorable niches in plants, animals, and the environment. Some studies have noted that *K. pneumoniae* can infect California sea lions and African green monkeys, causing invasive pneumonia (5, 6). It is also a common pathogen causing cow mastitis (7), and human clinical isolates share similar characteristics with strains from other sources (8). The World Health Organization recognizes extended-spectrum $\beta$-lactamase (ESBL)-producing and carbapenem-resistant *K. pneumoniae* as a critical public health threat (9). The problem of antibiotic resistance (AMR) caused by antibiotic drug abuse is becoming increasingly serious. At present, it represents an important challenge for global public health (10). It has been reported that the consumption of animal antibiotics exceeded 130,000 tons in 2013 (11), while the overall consumption of human antibiotic prescriptions expressed in defined daily doses increased by 65% from 2000 to 2015 (12). In agriculture, aquaculture, and intensive aquaculture, the use of antibiotics has exceeded four times that in the medical field (13). A large amount of rotten manure rich in antibiotics and antibiotic-resistant bacteria is used in agricultural production and may converge in surface water sources via runoff from fertilized land, leading to the retransmission and diffusion of drug-resistant strains or antibiotic resistance genes (ARGs) to humans or wild animals in contact with water sources (14). Mobile genetic elements such as plasmids, insertion sequences, transposons, and bacteriophages can mediate the horizontal transfer of antibiotic resistance genes among strains, attenuating the efficacy of antibiotics (15).

In previous studies, we found that red kangaroos in a zoo carried the multidrug-resistant *K. pneumoniae* isolate M297-1. Its genome and two plasmids carried $bla_{CTX}$, $bla_{TEM}$, *aph*, *aac*, *qnr*, and *fos*, which are closely related to the drug resistance phenotype and can endow *Escherichia coli* J53 with drug resistance through plasmid conjugate transfer between strains (16). In addition, a variety of drug resistance genes, including the carbapenemase genes $bla_{OXA-1}$ and $bla_{NDM-1}$, were found in the genome of *Proteus mirabilis* from the wild Malay pangolin and are closely related to a variety of mobile elements, such as the IS*26* insertion sequence and IntI1 integrase. In addition, the *E. coli* isolate M172-1 from a Malay pangolin sample carried the IncX1/IncX1 multireplicator plasmid pM172-1.3, which carried the complete IS*26*/IntI1/*arr-2*/*cmlA5*/$bla_{OXA-10}$/*ant(3')-IIa*/*dfrA14*/IS*26* structure, which may be formed by the copy fusion of two pM172-1.4 plasmids (IncX), giving the strain more extensive antibiotic resistance (17, 18). Several studies have shown that wild animal host-derived bacterial isolates carrying multiple drug resistance genes with related phenotypes, such as resistance to $\beta$-lactams, aminoglycosides, sulfonamides, and tetracyclines, have been widely spread all over the world (19–22).

There are obvious areas of overlap between wild birds and human activities, and their living environment is vulnerable to human activities. Their large range of activities and long flight distances make wild birds an important host and disseminator of strains with AMR (23). Approximately 5 billion migratory birds fly across continents every year, leading to the global prevalence of a variety of pathogens (24). A correlation analysis between ARG diversity and human density shows that in the presence of domestic animals, the diversity of ARGs carried by seed-eating birds increased with increasing human density. The bacterial gene community carried by birds is composed of strains derived from domestic livestock and poultry, human residents, and coexisting birds in the habitat. The genetics of the bacterial community carried by urban wild animals constitute a nonrandom process construction model (25). Human excreta (fecal sewage, wastewater, etc.) may contribute to the spread of ARGs to the wild even after treatment (26). Therefore, wild birds can be regarded as a potential repository of ARGs and antibiotic-resistant strains. A pair of *Salmonella enterica* isolates (SG17-135) with a phenotype of resistance to $\beta$-lactams, macrolides, aminoglycosides, sulfonamides, and other drugs and carrying the IncHI2 multidrug resistance (MDR) plasmid pSG17-135-HI2, which has a

**TABLE 1** Antimicrobial phenotypes of *K. pneumoniae* strains[a]

| Strain | Host | Nonsusceptible phenotype | Susceptible phenotype |
|---|---|---|---|
| S90-2 | Chukar | AMP, CXM, CZO, CRO, FEP, GEN, CHL, LVX, SXT | SAM, TZP, MEM, GEN, AMK, CHL, LVX, SXT, TGC |
| S141 | Red-breasted parakeet | AMP, CXM, CZO, GEN | TZP, MEM, GEN, AMK, CHL, LVX, SXT, TGC |
| M911-1 | Sun parakeet | AMP | CXM, CZO, CRO, FEP, SAM, TZP, MEM, GEN, AMK, CHL, LVX, SXT, TGC |
| S130-1 | Black-collared starling | NA | AMP, CXM, CZO, CRO, FEP, SAM, TZP, MEM, GEN, AMK, CHL, LVX, SXT, TGC |

[a]AMP, ampicillin; CXM, cefuroxime; CZO, cefazolin; CRO, ceftriaxone; FEP, cefepime; SAM, ampicillin-sulbactam; TZP, piperacillin-tazobactam; MEM, meropenem; GEN, gentamicin; AMK, amikacin; CHL, chloramphenicol; LVX, levofloxacin; SXT, trimethoprim-sulfamethoxazole; TGC, tigecycline; NA, not applicable.

complex resistance structure and carries 16 drug resistance genes (including $bla_{CTX-M-55}$), was identified from an Australian wild gull (*Chroicocephalus novaehollandiae*) (27). *E. coli* strains isolated from wild cattle egret (*Bubulcus ibis*) and white-faced tree duck (*Dendrocygna viduata*) in Ibadan, Nigeria, widely contain $bla_{CTX-M}$ family drug resistance genes, and the cattle egret isolates carry more ARGs and integrons, dominated by the IntI1 integrin gene (28). Wild migratory birds carry multidrug-resistant *E. coli*, and 43.7% of the 478 strains isolated from a sample were resistant *β*-lactam drugs, 22.6% were resistant to tetracycline drugs, and 73 strains were multidrug-resistant bacteria. The detected resistance genes mainly included $bla_{CTX-M}$, $bla_{TEM-1}$, *tet*(A), *tet*(B), *tet*(M), *sul1*, *sul2*, *sul3*, *cmlA*, and *floR*, indicating that the multidrug-resistant bacteria carried by the wild migratory birds came from the environment (29). For *Vibrio* species isolates carried by wild birds in the Danube delta of Romania, it was confirmed that 81.57% of the 76 isolates had multidrug resistance phenotypes. The main drugs associated with the phenotypes included penicillins, aminoglycosides, and macrolides. At the same time, the study also confirmed that the pathogenicity and drug resistance of *Vibrio* spp. carried by wild migratory birds were higher than those of strains carried by wild resident birds (30). In Spain, *Staphylococcus* strains carried by wild birds were resistant to methicillin, had an MDR phenotype, and carried the virulence genes *lukF*/S-Pv, *tst*, *eta*, *etb*, and *scn* (31). Neglect of the above problems will lead to the diffusion of MDR strains via the activities of bird hosts, and mobile genetic elements will further promote the transmission of drug resistance genes among humans, animals, and the environment. Therefore, exploring the drug resistance of strains carried by wild birds is of great biological significance for revealing the diffusion and transmission mode of AMR strains and establishing corresponding prevention and control measures.

In the present study, we isolated and identified four *K. pneumoniae* strains from wild Chukar partridge (*Alectoris chukar*), red-breasted parakeet (*Psittacula alexandri*), sun parakeet (*Aratinga solstitialis*), and black-collared starling (*Sturnus nigricollis*). Further whole-genome sequencing (WGS) and gene-phenotype association analysis were used to clarify the potential role and public health significance of wild birds as carriers and disseminators of MDR strains and antibiotic resistance genes.

## RESULTS

**Antimicrobial phenotype of *K. pneumoniae* strains.** Overall, the antimicrobial resistance phenotypes of the four *K. pneumoniae* strains were different. *K. pneumoniae* strain S90-2 showed resistance to 9 first-line antibiotics, namely, ampicillin, cefuroxime, cefazolin, ceftriaxone, cefepime, gentamicin, chloramphenicol, levofloxacin, and trimethoprim-sulfamethoxazole. *K. pneumoniae* strain S141 was highly resistant to ampicillin, cefuroxime, cefazolin, and gentamicin. *K. pneumoniae* M911-1 was resistant only to ampicillin; however, *K. pneumoniae* strain S130-1 was susceptible to 14 drugs (Table 1). All 4 *K. pneumoniae* strains were susceptible to piperacillin-tazobactam, the carbapenem meropenem, amikacin, and tigecycline (Table 1).

**Genomic structure and composition of *K. pneumoniae* strains.** The full length of the chromosome of *K. pneumoniae* S90-2 was 5,374,786 bp, belonging to sequence type 629 (ST629). The chromosome carried 12 gene islands and 30 antimicrobial resistance genes, including $bla_{CTX-M-14}$, *fosA6*, *aac(3)-IId*, and $bla_{SHV-11}$. It carried three plasmids, named pS90-2.1 [110,388 bp; IncFIB(pKPHS1)], pS90-2.2 [109,675 bp; IncFIA(HI1)/IncFII (K)], and pS90-2.3 (57,825 bp; IncR). pS90-2.3 carried 9 antimicrobial resistance genes,

**TABLE 2** Genomic information for *K. pneumoniae* isolates from wild birds

| Chromosome group or plasmid name of strain | ST | PlasmidFinder result | Movable resistance determinant(s) |
|---|---|---|---|
| Chr-S90-2 | ST629 | | $bla_{CTX-M-14}$, *fosA6*, *aac(3)-IId*, $bla_{SHV-11}$ |
| pS90-2.1 | | IncFIB(pKPHS1) | |
| pS90-2.2 | | IncFIA(HI1)/IncFII(K) | |
| pS90-2.3 | | IncR | *mphA*, *dfrA12*, *aadA2*, *qacEdelta1*, *sul1*, *tet*(A), *aph(3')-Ia*, *sul2*, *aph(3'')-Ib* |
| Chr-S141 | ST1662 | | *fosA5*, $bla_{SHV-217}$ |
| pS141.1 | | IncFIB(K)(pCAV1099-114)/repB | |
| pS141.2 | | IncFIB(pKPHS1) | |
| Chr-M911-1 | Novel | | $bla_{SHV-1}$, *fosA6* |
| pM911-1.1 | | Novel | *qnrS1*, $bla_{LAP-2}$, *tet*(A) |
| pM911-1.2 | | IncR/IncFII(pCTU2) | |
| pM911-1.3 | | Novel | |
| Chr-S130-1 | ST3753 | | $bla_{SHV-11}$, *fosA6* |
| pS130-1 | | IncFIB(K) | *tet*(A) |

including *mphA*, *dfrA12*, *aadA2*, *qacEdelta1*, *sul1*, *tet*(A), *aph(3')-Ia*, *sul2*, and *aph(3')-Ib*; however, pS90-2.1 and pS90-2.2 did not carry any antimicrobial resistance genes. The three plasmids contained 1 or 2 gene islands, and pS90-2.2 and pS90-2.3 also contained 1 prophage each (Table 2; see also Table S1 in the supplemental material).

The full length of chromosome S141 was 5,383,698 bp, belonging to ST1662. The genome carried 9 gene islands and 27 antimicrobial resistance genes, including *fosA5* and $bla_{SHV-217}$. S141 carried two plasmids, named pS141.1 [194,302 bp; IncFIB(K)(pCAV1099-114)/repB] and ps141.2 [112,160 bp; IncFIB(pKPHS1)]. pS141.1 contained five gene islands, and pS141.2 had one gene island. In addition, only plasmid pS141.1 carried a drug resistance-related efflux pump gene, *adeF* (Table 2 and Table S1). The full length of the chromosome of isolate M911-1 was 5,211,192 bp. It may be a new ST that was not retrieved in the MLST database. The genome carried 15 gene islands and 27 drug resistance genes, including mainly $bla_{SHV-1}$ and *fosA6*. M911-1 carried three plasmids, which were named pM911-1.1 (75,711 bp; novel), pM911-1.2 [85,824 bp; IncR/IncFII(pCTU2)], and pM911-1.3 (21,377 bp; novel). Plasmid pM911-1.1 carried three drug resistance genes, *qnrS1*, $bla_{LAP-2}$, and *tet*(A), and the other two plasmids did not carry drug resistance genes. pM911-1.1 contained two gene islands, pM911-1.2 had one gene island and one prophage, and pM911-1.3 had one prophage (Table 2 and Table S1). The full length of chromosome S130-1 was 5,249,027 bp, belonging to ST3753. The genome included 10 gene islands and 27 drug resistance genes, including $bla_{SHV-11}$ and *fosA6*. S130-1 carried a plasmid named pS130-1 [150,355 bp; IncFIB(K)], which carried only one drug resistance gene, i.e., *tet*(A), and three gene islands (Table 2 and Table S1).

**Homology analysis of the multidrug resistance plasmids pM911-1.1 and pS90-2.3.** Plasmid pM911-1.1 had a full length of 75,711 bp. It contained conjugate transfer regulatory genes, including *trbB*, *traY*, *traL*, *traI*, and *traJ*, on gene island pM911-1.1-GI-1 (bp 3 to 35601), but the *finO* gene and complete *tra-trb* gene cluster were not found. Therefore, it does not have the ability of conjugate transfer. There was also an IntI2 transposase gene on the gene island pM911-1.1-GI-1 (bp 40687 to 45449), but no antimicrobial resistance gene was found upstream or downstream of the IntI2 gene. In the PlasmidFinder database, the same incompatibility group type as that of the replication regulatory protein gene *repA* of the plasmid was not found, so we speculate that the plasmid belongs to a novel incompatibility group (Fig. 1A).

Further analysis showed that pM911-1.1 carried an MDR region (size, 13,104 bp; bp 50659 to 63762 bp [Fig. 2A]) containing three drug resistance genes, namely, *qnrS1*, $bla_{LAP-2}$, and *tet*(A), and their upstream and downstream transposable elements. This region was highly homologous to the *K. pneumoniae* plasmid pCRKP78R-4-tetA and *K. pneumoniae* plasmid p4_L39 in Zhejiang, China, the *Raoultella ornithinolytica* plasmid pWP8-W19-CRE-01_3 in Tokyo, Japan, and the *Klebsiella grimontii* plasmid p2481359-2 in Switzerland (Fig. 2B and Table S2). In addition, a 1,180-bp gene fragment (bp 54463 to 55643) in the pM911-1.1 MDR

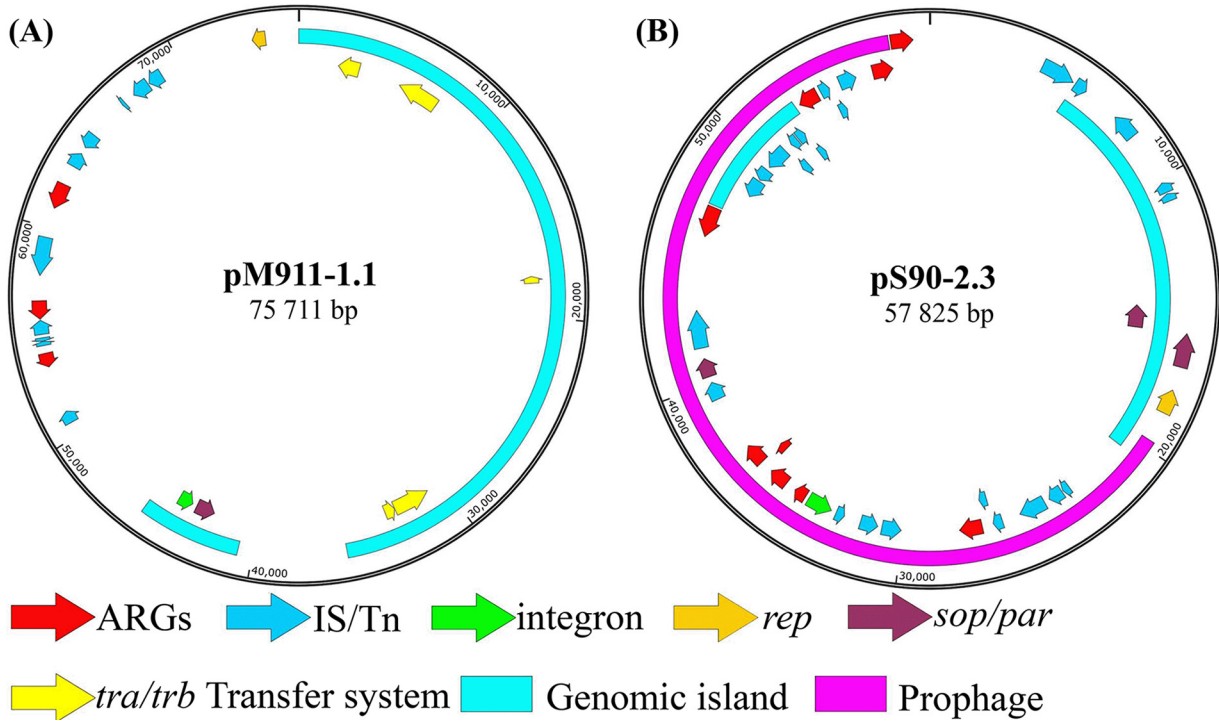

**FIG 1** Plasmid profiles of the multidrug resistance plasmids pM911-1.1 (A) and pS90-2.3 (B).

region contained ISEcl1 *insC21*, ISRso10 transposase gene, and ISMaq2 *insD*, while the corresponding region carried the complete IS*3* family transposase genes in the homologous plasmids pCRKP78R-4-tetA, p4_L39, pWP8-W19-CRE-013, and p2481359-2. In contrast, the tetracycline resistance gene *tet*(A) (bp 60925 to 62124) carried by plasmid pM911-1.1 was missing in the corresponding fragment of plasmid p2481359-2 (Fig. 2B).

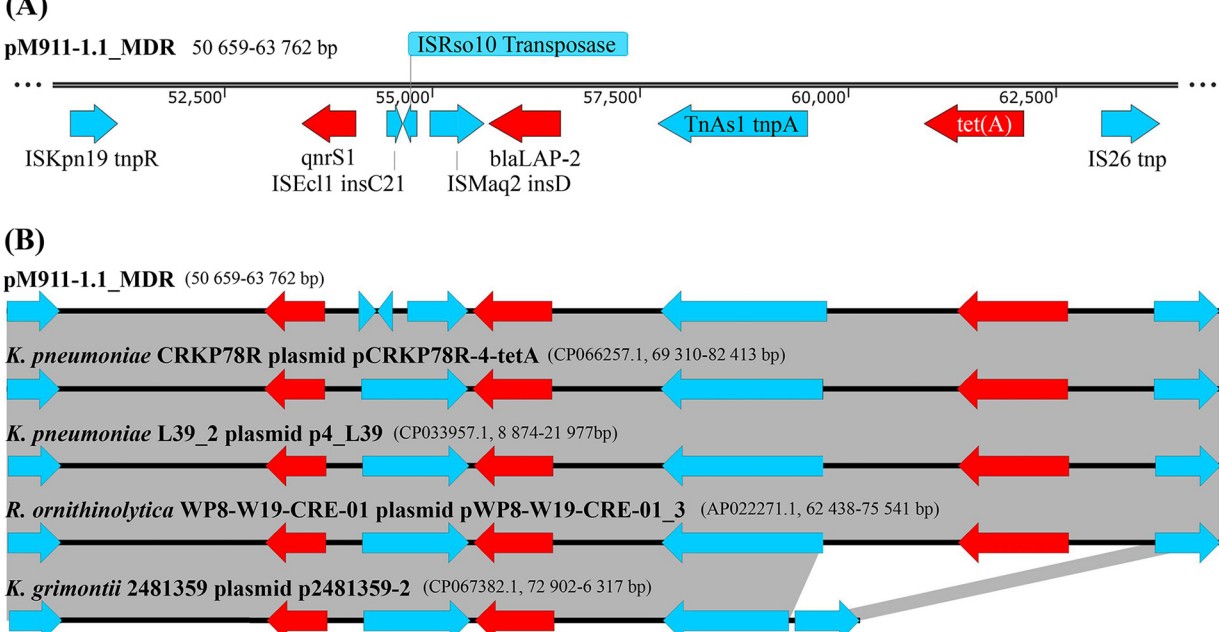

**FIG 2** Structure of the plasmid pM911-1.1 MDR region and its homologous fragment structure. (A) Schematic diagram of gene element combination in the pM911-1.1 MDR region; (B) comparison of the pM911-1.1 homologous fragment structure. Red fragment, drug resistance gene; blue fragment, transposase gene. The arrow direction represents the gene coding direction.

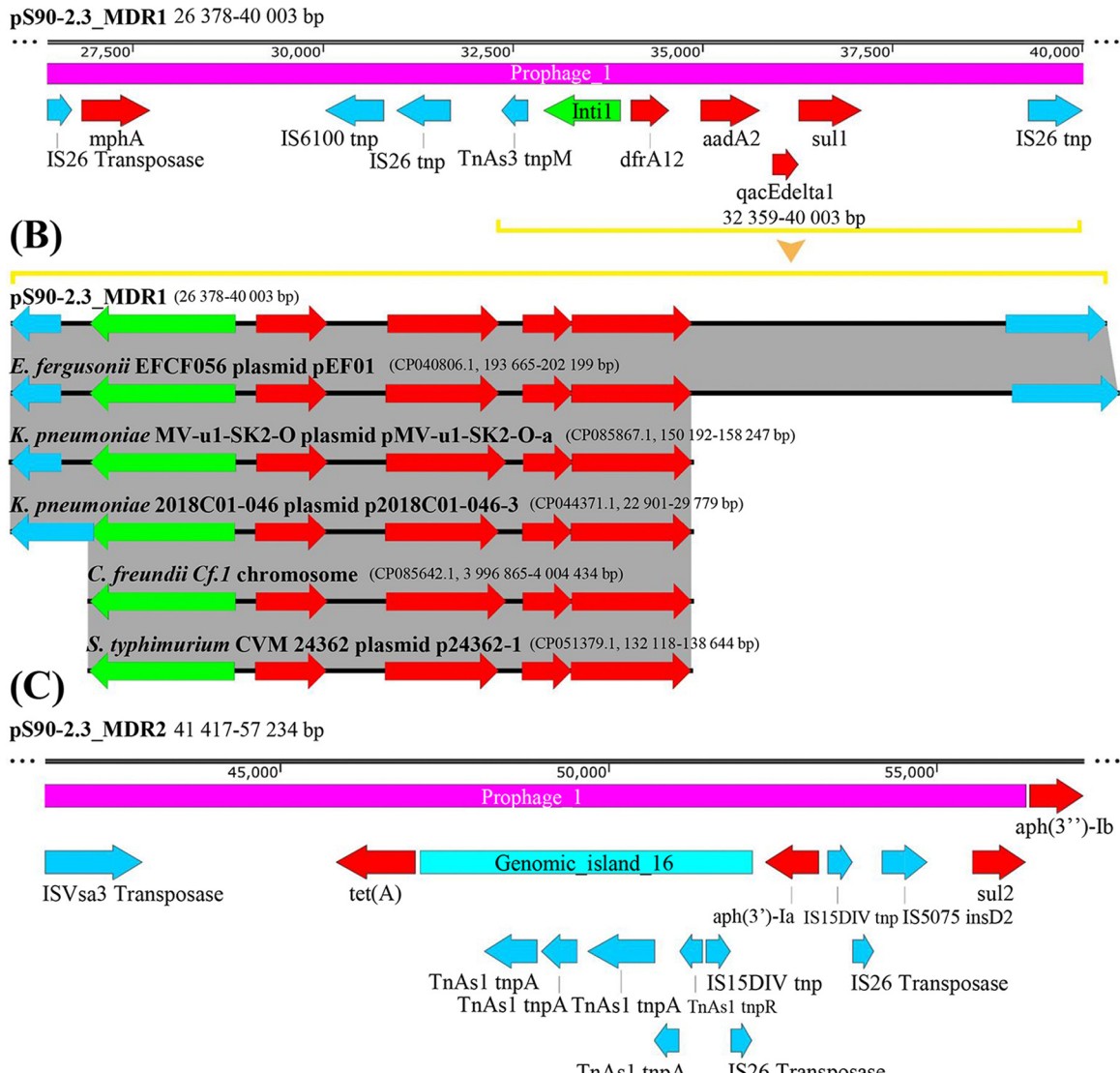

**FIG 3** Structure of the plasmid pS90-2.3 MDR region and its homologous fragment structure. (A) Schematic representation of gene element assembly in pS90-2.3 MDR1 region; (B) comparison of the pM911-1.1 homologous fragment structure; (C) schematic representation of gene element assembly in the pS90-2.3 MDR2 region. Red fragment, drug resistance gene; blue fragment, transposase gene; green fragment, integrase gene; purple fragment, prophage; cyan, gene island. The arrow direction represents the gene coding direction.

Plasmid pS90-2.3, with a total length of 57,825 bp, belonged to the IncR incompatibility group. The gene island pS90-2.3-GI-2 (bp 47134 to 52200) mainly contained Tn*As1*, IS*15*DIV, and IS*26* family transposase genes (Fig. 1B). In addition, 63.40% of the region of the plasmid was composed of a prophage structure (bp 19709 to 56370). Except for the phosphotransferase gene *aph(3″)-Ib* (bp 56431 to 57234), which mediates aminoglycoside drug tolerance, other antimicrobial resistance genes and an IntI1 integron structure were located in the prophage (Fig. 1B). According to the antimicrobial resistance gene carried by pS90-2.3, it was further divided into two MDR regions, including the MDR1 region (Fig. 3A; size, 13,626 bp; bp 26378 to 40003), carrying *mphA*, *dfrA12*, *aadA2*, *qacEdelta1*, and *sul1*, and the MDR2 region (Fig. 3C; size, 15,907 bp; bp 41417 to 57,234) carrying *tet*(A), *aph(3′)-ia*, *sul2*, and *aph(3″-ib)*. According to gene traceability analysis, a gene fragment (size, 7,644 bp; bp 32359 to 40003) in the MDR1 region was highly homologous with partial fragments of the *Escherichia fergusonii* plasmid pEF01 in Zhejiang,

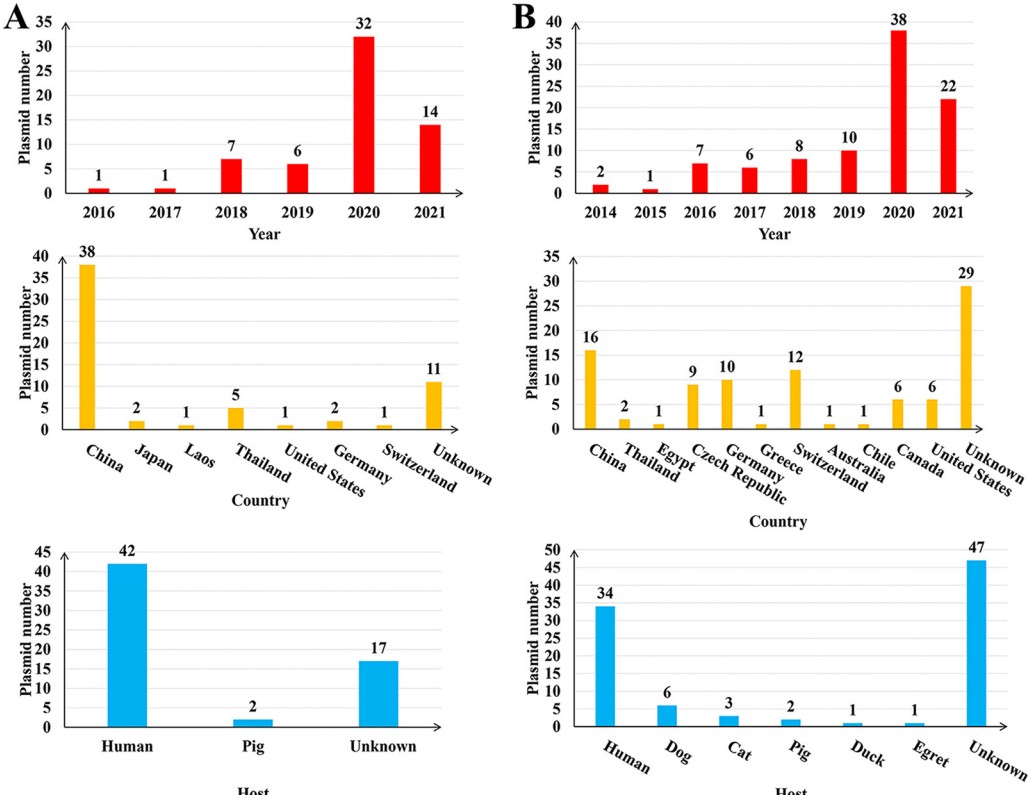

**FIG 4** Global distribution of the plasmid pM911-1.1 and pS90-2.3 homologs. (A) Time of isolation, country, and host information for the pM911-1.1 homologs; (B) time of isolation, country, and host information for the pS90-2.3 homologs.

China, *K. pneumoniae* plasmid p2018c01-046-3 in Taiwan, China, *K. pneumoniae* plasmid pMV-u1-SK2-O-a in Switzerland, and *Salmonella enterica* serovar Typhimurium plasmid p24362-1 in the United States. This structure was also highly homologous to some chromosomal sequences of *Citrobacter freundii* Cf.1, isolated from Guangxi, China (Fig. 3B and Table S2). Compared with the pS90-2.3 MDR1 structure, the plasmids pMV-u1-SK2-O-a and p2018C01-046-3 lacked the IS*26 tnp* fragment (bp 39299 to 40003), while *Citrobacter freundii* Cf.1 and plasmid p24362-1 lacked the Tn*As3 tnpM* fragment (bp 32359 to 32709) and IS*26 tnp* fragment (bp 39299 to 40003) (Fig. 3B). For pS90-2.3 MDR2, no fragments highly homologous to this region were found (Fig. 3C).

**Prevalence and distribution of multidrug resistance plasmids.** In the PLSDB (https://ccb-microbe.cs.uni-saarland.de/plsdb/), we screened a total of 155 similar plasmids. From 2016 to 2021, 61 of the plasmids screened were highly similar to pM911-1.1. Thirty-two of them (52.46%) were found in 2020, and 38 plasmids (62.30%) were mainly from China. In addition, 68.85% (42/61) came from human-infecting bacteria, and the others came from bacteria derived from pig hosts and those of unknown origin (Fig. 4A). Ninety-four plasmids were highly similar to pS90-2.3. From 2014 to 2021, most isolates were obtained in 2020 (42.43% [38/94]), and 17.02% (16/94) were from China. A total of 36.17% (34/94) were from human-infecting bacteria, and the others were from bacteria infecting dogs, cats, pigs, ducks, and egrets and those of unknown origin (Fig. 4B). pM911-1.1 and pS90-2.3 were homologous with some plasmids in the database (similarity ≥ 90%) in partial structural gene regions. Although the compositions were different, their main framework structure remained unchanged. These plasmids also had *K. pneumoniae* and *E. coli* as the main bacterial hosts (Fig. S1).

**Phylogenetic analysis based on the whole genome of *K. pneumoniae* carrying similar plasmids.** Among 155 similar plasmids, we screened 13 *K. pneumoniae* host bacteria carrying plasmids similar (similarity ≥ 95%) to pM911-1.1 and pS90-2.3, and the *K. pneumoniae* isolates S210-3, BS433-2, BM334-2 (isolated from a human) and M63-

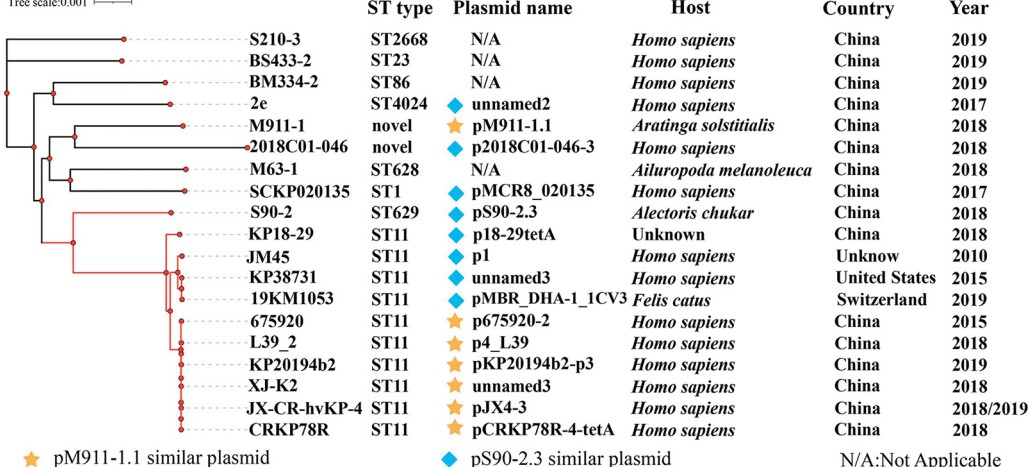

**FIG 5** Genome-wide evolutionary tree of the pM911-1.1 and pS90-2.3 homologous plasmids in the *K. pneumoniae* host and strains M911-1 and S90-2.3. Red branch, main clustering branch of similar plasmid hosts.

1 (isolated from a panda), isolated in our laboratory, were used to construct a phylogenetic tree based on the whole bacterial genome sequences. Phylogenetic analysis showed that 10 *K. pneumoniae* isolates belonged to ST11 and were located in the same branch. Among them, six strains were highly homologous with *K. pneumoniae* M911-1, and four strains were highly homologous with *K. pneumoniae* S90-2. *K. pneumoniae* 19KM1053 was isolated from domestic cats, *K. pneumoniae* KP18-29 is of unknown origin, and the other strains were from human samples (Fig. 5, red branch). In addition, *K. pneumoniae* 2e (ST4024) carrying a plasmid homologous to pS90-2.3 was in the same branch as *K. pneumoniae* BM334-2 (ST86) from human samples, *Klebsiella pneumoniae* 2018C01-046 (novel ST) was located in the same branch as M911-1 (novel ST), and SCKP020135 (ST1) was in the same branch as M63-1 (ST628; isolated from a panda) (Fig. 5). These *Klebsiella pneumoniae* isolates were mainly isolated from human samples in China from 2010 to 2019 (Fig. 5).

## DISCUSSION

In this study, we evaluated the antimicrobial resistance phenotype of four *K. pneumoniae* isolates from wild Chukar partridge, red-breasted parakeet, sun parakeet, and black-collared starling (Table 1). Studies have shown that wild birds can carry extended-spectrum $\beta$-lactamases (ESBLs) encoded by $bla_{CTX-M}$ and $bla_{SHV}$ from mainly highly pathogenic multidrug-resistant *Escherichia coli* (32–34), *K. pneumoniae* (35), *Pseudomonas* spp. (36), and *Campylobacter jejuni* (37) strains. Therefore, wild birds have become an important reservoir host for antimicrobial-resistant bacteria.

The genomes of all the *K. pneumoniae* isolates in this study carried the SHV resistance gene (Table 2). This SHV type shares approximately 68% homology with the TEM type based on the amino acid sequence, and their frame structures are similar, mainly in *E. coli*, *K. pneumoniae*, and *Pseudomonas aeruginosa* (38, 39). $bla_{SHV-1,}$ carried by S90-2 and M911-1, is the earliest-described hydrolase gene in the SHV ESBL family and is carried by plasmids or chromosomes of most *K. pneumoniae* strains and fewer *E. coli* strains (40). The $bla_{SHV-11}$ ESBL, carried by *K. pneumoniae* S130-1, was first found in a *K. pneumoniae* isolate from Switzerland (41), mediating $\beta$-lactam drug tolerance and involving nucleotide excision repair, mismatch repair, DNA replication, aromatic compound degradation, nitrogen metabolism, and amino acid metabolism regulation. $bla_{SHV-11}$ also induces DNA damage repair in coordination with *dnaJ*, *ligA*, *mutS*, *recA*, and *recF*, which is conducive to maintaining the integrity of the genome (42). $bla_{SHV-217}$, carried by S141, was found at a later stage, and there is little relevant information available for this gene. All the gene sequences of $bla_{SHV-217}$ are from *K. pneumoniae*. Therefore, it can mediate

the tolerance of penicillin and cephalosporins, which is consistent with the ampicillin resistance phenotype observed in this study (Table 1).

The chromosomes of S90-2, M911-1, and S130-1 carried the resistance gene *fosA6*, which mediates high-level fosfomycin resistance and was first found in the clinical strain *E. coli* producing the $bla_{CTX-M-2}$ enzyme in a hospital in the United States. Generally, *fosA6* is highly homologous to the genomic fragments of many strains of *K. pneumoniae* (43). Another homologous gene, *fosA5*, carried by S141 was found in ESBL isolate *E. coli* E265, which also exhibited high-level fosfomycin resistance. Gene tracing analysis showed that *fosA5* in the plasmid may have been derived from *K. pneumoniae* CG4 mediated by the insertion sequence IS*10* (44). In our study, it was also found that the chromosome of S90-2 carried the ESBL gene $bla_{CTX-M-14}$ and the acetyltransferase gene *aac(3)-lid*, which mediated β-lactam and aminoglycoside resistance (Table 1). Therefore, frequent gene exchange may occur among *Enterobacteriaceae* strains.

The lateral transfer of resistance genes is usually mediated by mobile genetic elements (such as insertion sequences, transposons, integrons, and prophages) (15). The MDR fragments carried by the multidrug resistance plasmids pM911-1.1 and pS90-2.3 were highly homologous with the fragments of chromosomes or plasmids from many *Enterobacteriaceae* strains (Fig. 2 and 3 and Table 2), indicating that there may be a variety of mechanisms for resistance gene transfer. The MDR region of pM911-1.1 was composed of *tnpR-qnrS1-insC21*-transposase-*insD-bla*$_{LAP-2}$-*tnpA-tet*(A)-*tnp*. There were transposase genes upstream and downstream of the resistance genes *qnrS1* and $bla_{LAP-2}$. The two resistance genes were separated by the transposase genes ISEcl1 *insC21*, ISRso10 transposase, and ISMaq2 *insD* and no longer shared Tn*3* transposase genes (Fig. 2A). Notably, there was only one complete IS*3* family transposase gene between the genes *qnrS1* and $bla_{LAP-2}$ in the homologous reference sequence (Fig. 2B). Therefore, it is speculated that the MDR of pM911-1.1 can be transferred horizontally mediated by ISEcl1, ISRso10, and ISMaq2 transposases, while the resistance gene cluster in the homologous reference plasmid (plasmids pCRKP78R-4-tetA, p4_L39, pWP8-W19-CRE-01_3, and p2481359-2 [Fig. 2]) only underwent a separate horizontal transfer of *qnrS1* or $bla_{LAP-2}$. The resistance gene *qnrS1* in the MDR fragment of pM911-1.1 was first found in a conjugated transfer plasmid from *Shigella flexneri*, which exhibits low-level fluoroquinolone resistance (45), and $bla_{LAP-2}$ is a β-lactamase resistance gene from *Enterobacter cloacae* (46). *tet*(A), a tetracycline efflux pump gene, was first found in many Gram-negative bacteria (47), but plasmid p2481359-2 lacked *tet*(A) (Fig. 2). Therefore, the MDR region may have evolved via multiple transposable or homologous recombination events.

According to the PlasmidFinder database, pM911-1.1 may represent a new type of plasmid incompatibility group, and the homology between the *rep* gene of pM911-1.1 and the *rep* gene of the IncFII(pCRY) incompatible plasmid was only 81.32%. IncFII plasmids are widely distributed all over the world and have become important vectors of ESBL genes such as $bla_{NDM}$, $bla_{SHV}$, $bla_{OXA}$, and $bla_{KPC}$, which mediate high levels of β-lactam resistance, including in *K. pneumoniae*, *E. coli*, and *Enterobacter cloacae* (48–51). Plasmid pCRY was first found in *Yersinia pestis* isolate 91001 (52), but another study confirmed that the similarity of the *repA* genes between the multidrug resistance plasmid pMET1 carried by the *K. pneumoniae* clinical isolate and plasmid pCRY was ≥95% (53), indicating that the plasmid carried by the bacterial host may have undergone natural evolution in the process of adapting to the environment.

The MDR1 fragment carried by plasmid ps90-2.3 was composed of IS*26* transposase-*mphA*-IS6100 *tnp*-IS*26* *tnp*-Tn*As3* *tnpM-intl1-dfrA12-aadA2-qacEdelta1-sul1*-IS*26* *tnp* (Fig. 3A), which mediates resistance to macrolides, trimethoprim, aminoglycosides, and sulfonamides (54–58). Its core structure is Tn*As3* *tnpM-intl1-dfrA12-aadA2-qacEdelta1-sul1*-IS26 *tnp*, in which an integron structure with a size of 4,199 bp (*intl1-dfrA12-aadA2-qacEdelta1-sul1*) exists. It has been reported that the classic structure of the integron mainly exists in the chromosomes of *E. coli* and *K. pneumoniae* (similarity, 100%); however, its upstream and downstream regions lack transposase genes (Fig. 3B). The *tnpM* and *tnp* genes are

located upstream and downstream of the integron structure in pS90-2.3; thus, this fragment constitutes both the integron structure and the transposon structure. Therefore, these results suggest that the transposase gene promotes the transfer of integron fragments to plasmids and further enhances the transmission of drug resistance genes (59, 60).

The MDR2 region of pS90-2.3 contains four drug resistance genes, which mainly mediate the tolerance of tetracyclines, aminoglycosides, and sulfonamides. In addition, there are 11 transposase genes in this region, of which 7 are located on the gene island (Fig. 3C), indicating that MDR2 may be formed by transposon-mediated multiple gene horizontal transfer events. Notably, 63.40% of the pS90-2.3 plasmid is a prophage structure (bp 19709 to 56370), which is 100% homologous to the P1-like phage RCS47 (GenBank accession number NC_042128.1) found in *E. coli* 725 (serotype O8:H19). RCS47 carries only one antibiotic resistance gene, $bla_{SHV-2}$, and there are multiple transposase genes, IS*26*, IS*5*, and IS*1*, upstream and downstream of this gene (61). In this study, plasmid pS90-2.3 carried 9 drug resistance genes, among which 8 were provided by the prophage (Fig. 1B and Fig. 3), except for *aph(3')-Ib* (bp 56431 to 57234). Similar to phage RCS47, this plasmid also carried multiple transposase genes. These transposases provide conditions for the plasmid to obtain drug resistance genes and integrate phage-carrying drug resistance genes. Plasmid pS90-2.3 belongs to the IncR type, which was first discovered in 2009, and does not have the capability of conjugation and transfer (62). Among clinical isolates of *K. pneumoniae*, the IncR plasmid mainly carries the drug resistance gene $bla_{KPC-2}$, $bla_{DHA-1}$, $bla_{NDM-1}$, $bla_{VIM-1}$, *qnrS1*, or *armA* (63–65). IncR plasmids can coexist with many types of plasmids, such as IncC, IncN, IncHI, and IncFII, and the drug resistance genes carried by them can be transferred laterally through transposition or plasmid recombination events, thus promoting the diffusion of drug resistance genes among bacterial species (66).

There are 155 plasmids highly homologous to pM911-1.1 and pS90-2.3, distributed mainly in China and its surrounding countries and to a lesser extent in Europe and the Americas (Fig. 4). The plasmid homologous to pM911-1.1 was reported in Germany, Thailand, Laos, and China before 2020; it was mainly reported in China and Japan in 2020 and mainly in China, Thailand, the United States, and Switzerland in 2021. Similarly, plasmids homologous to pS90-2.3 were primarily found in the United States, Canada, Germany, Greece, Egypt, Thailand, and China before 2020, isolated in Switzerland, Germany, Chile, Australia, Canada, and China in 2020, and reported in the Czech Republic, China, the United States, and Switzerland in 2021. Therefore, the above two plasmids closely related to the host bacteria of humans have been widely spread and distributed in China and may be gradually becoming the dominant plasmid type.

In this study, the habitats of sun parakeet and Chukar partridge carrying drug-resistant bacteria and migratory birds overlapped with human colonies, and the migratory behavior of wild birds may further increase the risk of spread of drug-resistant strains and drug resistance plasmids. The host bacteria of the drug resistance plasmid were mainly ST11 *K. pneumoniae* (Fig. 5, red branch). In the last 10 years, ST11 *K. pneumoniae* has become a popular dominant clone in China (67). Among ST11 *K. pneumoniae* isolates, ESBL-carrying *K. pneumoniae* isolates that widely appear are closely associated with IncFII-like plasmids (68). Under the mediation of the insertion sequence IS*26*, a variety of ESBL genes can coexist in IncF/IncR-type plasmids carried by ST11 *K. pneumoniae* (69). The plasmids carried by ST11 ESBL-producing *K. pneumoniae* isolates prevalent in Asia are mainly recombinant plasmids (70). The isolates M911-1 and S90-2 in this study were not ST11 *K. pneumoniae* strains, and the host bacteria of homologous plasmids shared a close genetic relationship with other-ST non-drug-resistant *K. pneumoniae* strains (Fig. 5). Therefore, ST11 *K. pneumoniae* may be an important host of multidrug resistance plasmids, but multidrug resistance plasmids can also spread in a variety of different host bacteria.

**Conclusion.** In conclusion, *K. pneumoniae* carried by wild birds can carry multidrug resistance plasmids, which are closely related to human clinical isolates. These plasmids obtain drug resistance genes through a variety of mobile elements and endow the strains with multidrug resistance phenotypes. Therefore, wild birds may become a potential repository of drug-resistant bacteria with clinically significant drug resistance genes, which further increases the possibility of AMR horizontal transfer among humans, animals, and the environment, thus constituting a hidden danger to public safety.

## MATERIALS AND METHODS

**Strain isolation.** *K. pneumoniae* strains were isolated from feces of smuggled wild birds, including Chukar partridge (*Alectoris chukar*), red-breasted parakeet (*Psittacula alexandri*), sun parakeet (*Aratinga solstitialis*), and black-collared starling (*Sturnus nigricollis*). *K. pneumoniae* S90-2 and M911-1 were isolated from a Chukar partridge and sun parakeet, respectively, in Guangzhou, Guangdong Province, China, and *K. pneumoniae* S141 and S130-1 were isolated from a red-breasted parakeet and black-collared starling, respectively, in Heyuan, Guangxi Province. The bacterial cells were cultured on the surface of a MacConkey agar plate (Beijing Sanyao Science & Technology Development Co., Beijing, China) at 35.2℃ for 18 h. A single colony was selected and cultured in Mueller-Hinton broth (MHB) liquid medium at 35.2℃ for 18 h to obtain the enrichment solution for subsequent experiments.

**Antibiotic susceptibility testing.** In accordance with the guidelines issued by the CLSI in 2021 (71) and the EU standard for antimicrobial susceptibility testing published by EUCAST in 2021 (72), ampicillin, cefuroxime, ceftriaxone, cefepime, ampicillin-sulbactam, piperacillin-tazobactam, meropenem, gentamicin, amikacin, and chloramphenicol were used for antibiotic susceptibility testing at the MIC. The reference strain *E. coli* ATCC 25922 was used as the quality control strain.

**Whole-genome sequencing.** Whole-genome sequencing was performed using the Nanopore sequencing platform (Biomarker Technologies, China) (73, 74). High-quality genomic DNA was extracted, and quality inspection was performed by a Nanodrop, a Qubit, and 0.35% agarose gel electrophoresis. The Bluepippin automatic nucleic acid recovery system recovers large pieces of DNA. Library construction was performed with an sqk-lsk109 ligation kit (including DNA damage repair and terminal repair, junction connection, magnetic bead purification, and Qubit library quantification), and the library was subjected to sequencing. After obtaining the data, the subreads with low quality and those that were too short were filtered, and canu v1.5 software was used to reassemble the filtered subreads from scratch. The draft genome was assembled with Pilon software. The genomic DNA library was constructed, and whole-genome sequencing was performed, with an estimated size of 6 Mbp. The sequencing depth was ≥100×, with 0 gaps.

**Genome annotation.** Gene elements were annotated by using the NR, UniProt, COG, and KEGG databases, the transposon registry (https://transposon.lstmed.ac.uk/), the insert sequence database ISfinder (https://www-is.biotoul.fr/index.php), and the Integrall database (http://integrall.bio.ua.pt/). The incompatible group types of plasmids were analyzed by using the PlasmidFinder database (https://bitbucket.org/genomicepidemiology/plasmidfinder). The Comprehensive Antibiotic Resistance Database (CARD; https://card.mcmaster.ca/home) was used to annotate the genes related to drug resistance. Furthermore, seven conserved housekeeping genes (*rpoB*, *gapA*, *mdH*, *pgi*, *phoE*, *infB*, and *tonB*) were analyzed for multilocus sequence types (MLST) by using the *K. pneumoniae* MLST database of BIGSDB Pasteur (https://bigsdb.pasteur.fr/). The species of the strains were identified by using the ribosomal MLST database (rMLST; https://pubmlst.org/species-id).

**Structural analysis of multidrug resistance plasmids.** According to the annotation results of the CARD database, the multidrug resistance plasmids pM911-1.1 and pS90-2.3 were selected, and the plasmid map was drawn by SNAPGENE software (from Insightful Science; available at https://www.snapgene.com/). The NCBI database was used for BLAST analysis of plasmid drug resistance gene fragments, and EasyFig (75) was used to compare the differences in drug resistance genes and their upstream and downstream drug resistance gene-related elements (similarity ≥ 95%). The plasmid database PLSDB (https://ccb-microbe.cs.uni-saarland.de/plsdb/) was used to search for similar plasmids (the search conditions were limited to max.$P$ value = 0, max.distance = 0.04, per. Ident ≥ 60%), and their isolation year, country, and host information were collected. Some similar plasmids (similarity ≥ 90%) were selected for plasmid genome difference comparison, which was performed by mauve software (76).

**Construction of the whole-genome phylogenetic tree of *K. pneumoniae*.** According to the results for similar plasmids retrieved from the PLSDB (similarity ≥ 95%), *K. pneumoniae* isolates of some plasmids were selected for MLST, the bacterial genome-wide phylogenetic tree was constructed by using the tool REALPHY 1.13 (77), and the ST, host, country, isolation date, and other information for the bacterial isolates were collected for epidemic distribution analysis.

**Data availability.** All accession numbers for bacterial genomes or plasmids related to the paper were deposited in the GenBank database (Tables S1 and S2). All the data are available in the main text or supplemental material.

## SUPPLEMENTAL MATERIAL

Supplemental material is available online only.

**SUPPLEMENTAL FILE 1**, PDF file, 0.4 MB.

## ACKNOWLEDGMENTS

This study was funded by the Introduction of Leading Talents Program of the Guangdong Academy of Sciences (no. 2016GDASRC-0205) and Open project of Beijing Key Laboratory of captive wildlife technology in Beijing Zoo (ZDK202105).

The study was designed and supervised by C.W. and X.W.; J.Z., F.J., and M.W. isolated and identified bacterial isolates; G.D., J.Q., R.Z., and X.W. collected all the samples; X.W. and B.W. analyzed the data; J.Z., X.W., and C.W. prepared the original draft; and X.W., J.Z., and C.W. reviewed and edited the manuscript.

We declare that we have no competing interests.

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
