## [Reviewer comments · Microbiology Spectrum]

Microbiology Spectrum

Genomic characteristics and molecular epidemiology of multidrug resistant *Klebsiella pneumoniae* carried by wild birds

Xue Wang, Jianan Zhao, Fang Ji, Meng Wang, Bin Wu, Jianhua Qin, Guoying Dong, Ruili Zhao, and Chengmin Wang

Corresponding Author(s): Chengmin Wang, Guangdong Academy of Sciences

Review Timeline:

Submission Date:	July 13, 2022
Editorial Decision:	October 17, 2022
Revision Received:	December 6, 2022
Editorial Decision:	December 29, 2022
Revision Received:	January 31, 2023
Accepted:	February 4, 2023

Editor: Diyan Li

Reviewer(s): Disclosure of reviewer identity is with reference to reviewer comments included in decision letter(s). The following individuals involved in review of your submission have agreed to reveal their identity: Hua Zhong Chen (Reviewer #2); Ling Wang (Reviewer #6); Yung-Fu Chang (Reviewer #7)

Transaction Report:

DOI: <https://doi.org/10.1128/spectrum.02691-22>

October 17, 2022

Prof. Chengmin Wang
Institute of Zoology, Guangdong Academy of Science
Guangdong Key Laboratory of Animal Conservation and Resource Utilization
Guangzhou, Guangdong
China

Re: Spectrum02691-22 (Genomic characteristics and molecular epidemiology of multidrug resistant *Klebsiella pneumoniae* carried by wild birds)

Dear Prof. Chengmin Wang:

Link Not Available

Sincerely,

Diyang Li

Journals Department
Reviewer comments:

Reviewer #1 (Comments for the Author):

The study investigate the genomic features of pathogen bacteria from the wild birds. I have some major concerns:

- (1) the title is so large. This study only focus on several species, and 'wild birds' in the title is over-stated.
- (2) what's reason for choose these species?
- (3) the samples size is small, and the author only study one bacteria. Thus, the novelty and importance is low.
- (4) as I know, now more and more study on the wild birds focus on the virus, which may have the negative effects on the human health. Thus, I don't think this study will bring the interesting for the readers.
- (5) the authors explore the genomic features on this bacteria, and this study may be fit some microbial journal on genomic

studies.

Reviewer #2 (Comments for the Author):

Manuscript ID # Spectrum02691-22 entitled "Genomic characteristics and molecular epidemiology of multidrug resistant *Klebsiella pneumoniae* carried by wild birds" by Xue Wang et al. reports MDR *Klebsiella pneumoniae* were isolated from fresh feces of captured wild bird and showed drug-resistant phenotypes conferred mainly by multidrug-resistant plasmids.

The major comments:

The problem of antibiotic resistance (AMR) caused by antibiotic drug abuse is becoming more and more serious. At present, it has become an important challenge of global public health. The multidrug-resistant *Klebsiella pneumoniae* isolates carried by wild birds had drug-resistant phenotypes conferred mainly by multidrug-resistant plasmids. Wild bird habitats overlap to a greater extent with human and livestock habitats, and further increase the potential for horizontal transfer of multi-drug resistant bacteria between human-animal-environment. This study is very importance to explore the relationship between wild birds and the transmission of multi-drug resistant strain.

Some minor comments:

Line 136, "DNA;" revised to "DNA."

Line 138, ";" revised to "and"

Line 185, "5,374,786kb" should be "5,374,786 b"

Line 211, "5711bp" should be "5,711bp"

Table 1 and Table S1, bird host name should be revised to "English name"

I recommend acceptance after the above minor modifications.

Reviewer #6 (Comments for the Author):

Interesting/relevant; but suggest to enlarge the number and species of animals

Reviewer #7 (Comments for the Author):

This is an interesting manuscript that may be interesting to those who are working in this field. There are multiple typos that need to be fixed. Please see the enclosed marked manuscript.

Staff Comments:

Preparing Revision Guidelines

Please return the manuscript within 60 days; if you cannot complete the modification within this time period, please contact me. If you do not wish to modify the manuscript and prefer to submit it to another journal, please notify me of your decision immediately so that the manuscript may be formally withdrawn from consideration by Microbiology Spectrum.

If your manuscript is accepted for publication, you will be contacted separately about payment when the proofs are issued;

please follow the instructions in that e-mail. Arrangements for payment must be made before your article is published. For a complete list of **Publication Fees**, including supplemental material costs, please visit our website.

**Genomic characteristics and molecular epidemiology of multi-drug**
**resistant *Klebsiella pneumoniae* carried by wild birds**

**Authors:** Xue Wang^{1*}, Jianan Zhao^{1*}, Fang Ji¹, Meng Wang^{1,2}, Bin Wu¹, Jianhua Qin², Guoying Dong³,
Ruiqi Zhao⁴, Chengmin Wang^{1#}

**Affiliations:**

1. Guangdong Key Laboratory of Animal Conservation and Resource Utilization, Institute of
Zoology, Guangdong Academy of Science, Guangzhou 510260, Guangdong Province, China

2. College of Veterinary Medicine, Agricultural University of Hebei, Baoding 071001, Hebei
Province, China

3. College of Global Change and Earth System Science, Beijing Normal University, Beijing
100875, China.

4. College of Animal Science and Veterinary Medicine, Tianjin Agricultural University, Tianjin,
China,

*Equally to this work.

#Corresponding author. E-mail: wangchm@giz.gd.cn

**Abstract**

This study aimed to explore the relationship between wild birds and the transmission of multi-drug
resistant strains. *Klebsiella pneumoniae* were isolated from fresh feces of captured wild birds and were
assessed by micro-broth dilution method and comparative genomics. Four *Klebsiella pneumoniae*
isolates showed different resistance phenotypes, S90-2 and S141 ~~was~~ were all resistant to ampicillin,
cefuroxime, cefazolin, but M911-1 and S130-1 was sensitive to most of the 13 antibiotics. S90-2 belongs
to ST629₂ and its genome carries 30 resistance genes₂ including bla_{CTX-M-14} and bla_{SHV-11}, while its plasmid
pS90-2.3 (IncR) carries *qacEdelta1*, *sul1* and *aph(3')-Ib* etc. S141 belongs to ST1662 and its genome
carries a total of 27 resistance genes₂ including bla_{SHV-217}. M911-1 is a new ST type, carrying bla_{SHV-1} and
fosA6, and its plasmid pM911-1.1 (novel) carries *qnrS1*, *bla_{LAP-2}* and *tet(A)*. S130-1 belongs to ST3753,
carrying bla_{SHV-11} and fosA6₂ and its plasmid pS130-1 (IncFIB(K)) carries only one resistance gene tet(A).
The pM911-1.1 and pS90-2.3 do not have conjugative transfer ability, but their resistance gene fragments
are derived from multiple homologous Enterobacteriaceae strains chromosomes or plasmids, and the

Formatted: Font: Italic

Formatted: Font: Italic

Formatted: Font: Italic

Formatted: Font: Italic

Formatted: Font: Italic

Formatted: Font: Italic

Formatted: Font: Italic, Underline

formation of resistance gene fragments (MDR region) involves interactions between multiple mobile
element genes, resulting in a complex and diverse resistance plasmid structure. The homologous
plasmids related to pM911-1.1 and pS90-2.3 were mainly from human bacteria hosts in China, whose are
~~*Klebsiella-K. pneumoniae*~~ and *Escherichia coli*. The multidrug-resistant ~~*Klebsiella-K. pneumoniae*~~ isolates
carried by wild birds in this study had drug-resistant phenotypes conferred ~~mainly-primarily~~ by multidrug-resistant
plasmids which were closely related to human bacteria hosts.

Formatted: Font: Italic

Formatted: Font: Italic

**Keywords:** Wild birds; ~~Multi drug~~ **Multidrug** resistant bacteria; *Klebsiella pneumoniae*; Drug-resistant
plasmids

**Introduction**

The problem of antibiotic resistance (AMR) caused by antibiotic drug abuse is becoming more and
more serious. At present, it has become an important challenge ~~of-for~~ global public health (Burnham et
al.,2017). It is reported that the consumption of animal antibiotics exceeded 130000 tons in 2013 (Van
Boeckel et al.,2017), while the overall consumption of human antibiotic prescriptions expressed in
defined daily dose increased by 65% ~~during-from~~ 2000 to 2015 (Klein et al.,2018). In ~~the-fields-of~~ agriculture,
aquaculture, and intensive aquaculture, the use of antibiotics has even reached more than four times that
in the medical field (Smith et al.,2014). A large number of rotten manure rich in antibiotics and antibiotic
resistant bacteria are used in agricultural production, and may converge to surface water sources by
fertilizing land runoff, which will lead to the ~~re-re~~ transmission and diffusion of drug-resistant strains or
drug-resistant genes (ARGs) to humans or wild animals in contact with water sources (Laxminarayan et
al.,2013). Mobile gene elements such as plasmids, insertion sequences, transposons and bacteriophages
can mediate the horizontal transfer of antibiotic resistance genes among strains, making the efficacy of
antibiotics increasingly attenuated (Partridge et al.,2018). In previous studies, we found that red
kangaroos in zoo circle carry ~~multi-multi~~ drug resistant *Klebsiella-K. pneumoniae* isolate M297-1. Its genome and
two plasmids carry *bla*_{CTX}, *bla*_{TEM}, *aph*, *aac*, *qnr* and *fos* closely related to drug resistance phenotype,
and can endow ~~*Escherichia-E. coli*~~ J53 with drug resistance through plasmid conjugate transfer between
strains (Wang et al.,2020). A variety of drug resistance genes, including carbapenemase *bla*_{OXA-1} and
*bla*_{NDM-1} were found in the genome of *Proteus mirabilis* from wild Malay pangolin, which are closely
related to a variety of mobile elements such as IS26 insertion sequence and IntI1 integrase. In addition,

the *Escherichia-E. coli* isolate M172-1 of Malay pangolin sample carries the IncX1/IncX1 multireplicator
plasmid pM172-1.3, which carries the complete *Is26/Int1/arr-2/cmlA5/bla_{OXA-10}/ant(3'')*-IIa/dfrA14/Is26
structure, which may be formed by the copy fusion of two pM172-1.4 plasmids (IncX), giving the strain
more extensive antibiotic resistance (Kang et al.,2021; Ji et al.,2022). More studies have shown that wild
animal host bacterial isolates are carried multiple drug resistance genes with related phenotypes such as
β - lactams, aminoglycosides, sulfonamides and tetracyclines have been widely spread all over the world
(Stokes et al.,2011; Eiamphungporn et al.,2018; Bush et al.,2020; Chang et al.,2015).

There are obvious overlapping areas between wild birds and human activities, and the living
environment is vulnerable to human activities. Its large range of activities and long flight distance make
it an important host and disseminator of AMR strains (Wang et al.,2017). About 5 billion migratory birds
fly across continents every year, leading to the global prevalence of a variety of pathogens (Guenther et
al.,2012). The correlation analysis between ARGs diversity and human density shows that when there are
domestic animals, the diversity of ARGs carried by Valley eating birds increases with the increase of
human density. The bacterial gene community carried by birds is composed ~~of the combination of~~
domestic livestock and poultry, human ~~residence-residents~~, and coexisting bird habitats. The bacterial genetic
community carried by urban wild animals forms a non-random process construction model (Hassell et
al.,2019), Human excreta (fecal sewage, wastewater, etc.) may contribute to the spread of ARGs to the
wild even after treatment (Marcelino et a.,2019). Therefore, wild birds can be regarded as a potential
repository of ARGs and antibiotic resistant strains. Australian wild gull (*Chrococephalus*
*novaehollandiae*), a pair of *Salmonella enterica* isolates SG17-135 with a phenotype of β -lactams,
macrolides, aminoglycosides, sulfonamides and other drugs ~~carries-carrying~~ IncHI2 multi-drug resistance (MDR)
plasmid pSG17-135-HI2 with a complex resistance structure, which carries 16 drug resistance genes
including *bla_{CTX-M-55}* (Cummins et al.,2020). *Escherichia-E. coli* isolated from the wild cattle egret
(*Bubulcus ibis*) and the white-faced tree duck (*Dendrocygna viduata*) in Ibadan, Nigeria, widely contains
the *bla_{CTX-M}* family drug resistance gene, while the cattle egret isolates carry more ARGs and integrons
dominated by the IntI1 integrin gene (Fashae et al.,2021). Wild migratory birds carried ~~multi-multi~~-drug
resistant *Escherichia-E. coli*, and 43.7% of the 478 strains isolated from the sample were resistant β - Among
the lactam drugs, 22.6% were resistant to tetracycline drugs, and 73 strains were ~~multi-drug-multidrug~~-resistant
bacteria. The detected resistance genes mainly included *bla_{CTX-M}*, *bla_{TEM-1}*, *tet(A)*, *tet(B)*, *tet(M)*, *suI1*,
*suI2*, *suI3*, *cmlA* and *floR*, indicating that the ~~multi-drug-multidrug~~-resistant bacteria carried by wild migratory birds
came from the environment (Yuan et al.,2021). The *Vibrio spp.* isolates carried by wild birds in the
Danube delta of Romania confirmed that 81.57% of the 76 isolates had multiple drug resistance
phenotypes. The main resistant drugs included penicillins, aminoglycosides, and macrolides. At the same

time, it also confirmed that the pathogenicity and drug resistance of *Vibrio* spp carried by wild migratory
birds were higher than those of wild resident birds (Páll et al.,2021). In Spain, *Staphylococcus* carried by
wild birds is resistant to methicillin and has MDR phenotype, and carries virulence genes *lukF* / S-Pv, *tst*,
*eta*, *etb*, *etd* and *scn* (Ruiz-Ripa et al.,2020). The ~~above-above~~-neglected problems will lead to the diffusion of
MDR strains with the activities of bird hosts, and mobile gene elements will further promote the
transmission of drug-resistant genes among humans, animals and the environment. Therefore, exploring
the drug resistance of strains carried by wild birds is of great biological significance to reveal the
diffusion and transmission mode of AMRS and establish corresponding prevention and control measures.

In our present study, we isolated and identified four ~~*Klebsiella-K. pneumoniae*~~ strains from wild
Chukar Partridge (*Alectoris chukar*), Red-breasted Parakeet (*Psittacula alexandri*), Sun Parakeet
(*Aratinga solstitialis*) and Black-collared Starling (*Sturnus nigricollis*). Further whole genome
sequencing (WGS) technology and ~~gene-gene~~-phenotype association analysis was used to clarify the potential
role and public health significance of wild birds as carriers and disseminators of MDR strains and
antibiotic resistance genes.

**Materials and methods**

***Strains isolation***

~~*Klebsiella-K. pneumoniae*~~ strains were isolated from smuggling wild birds, including Chukar Partridge
(*Alectoris chukar*), Red-breasted Parakeet (*Psittacula alexandri*), Sun Parakeet (*Aratinga solstitialis*) and
Black-collared Starling (*Sturnus nigricollis*). ~~*Klebsiella-K. pneumoniae*~~ S90-2 and M911-1 were from
Chukar Partridge and Sun Parakeet in Guangzhou, Guangdong Province, and ~~*Klebsiella-K. pneumoniae*~~
S141 and S130-1 were from Red-breasted Parakeet and Black-collared Starling in Heyuan, Guangxi
province, respectively. The bacterial cells were cultured on the surface of ~~the~~ MacConkey agar plate (Beijing
Sanyao Science & Technology Development Co, Beijing, China) at 35.2 °C for 18 hours. A single colony
was selected and cultured in Mueller Hinton broth (MHB) liquid medium at 35.2 °C for 18 hours to
obtain the enrichment solution for subsequent experiments.

***Antibiotic Susceptibility Test***

According to the standard for antimicrobial susceptibility test issued by CLSI in 2021 (CLSI, 31st ed)
and the EU standard for antimicrobial susceptibility test ~~issued-published~~ by EUCAST in 2021 (version 10.0, 2021).
Ampicillin, Cefuroxime, Ceftriaxone, Cefepime, Ampicillin/Sulbactam, Piperacillin/Tazobactam,
Meropenem, Gentamicin, Amikacin, and Chloramphenicol were used ~~to-for the~~ Antibiotic susceptibility test by
the Minimum Inhibitory Concentration (MIC). The reference strain ~~*Escherichia-E. coli*~~ ATCC25922 was
used as the quality control strain.

**Whole Genome Sequencing**

Whole genome sequencing using nanopore sequencing platform (biomarker technologies, China)
(Loman et al.,2015; Ashton et al.,2015). High quality genomic DNA was extracted and quality inspection
was performed by Nanodrop, Qubit, and 0.35% agarose gel electrophoresis. Bluepippin automatic nucleic
acid recovery system recovers large pieces of DNA; Library construction (sqk-lsk109 ligation kit,
including DNA damage repair and terminal repair, junction connection, magnetic bead purification, and
qubit library quantification); Computer sequencing. After obtaining the data, filter the subreads with low
quality and too short length, and use the software canu v1.5 to reassemble (company, city, country?) the filtered subreads
from scratch,
correct the assembled draft genome with the software Pilon, construct the genomic DNA library, and
complete the sequencing of the bacteria with an estimated size of 6m. The sequencing depth is $\geq 100\times$, 0
gap.

**Genome Annotation**

Gene elements are annotated by using NR, UniProt, COG and KEGG, transposon registry TN number
registry(<https://transposon.lstmed.ac.uk/>), insert sequence database ISfinder (<https://www-is.biotoul.fr/index.php>) and integral sub database(<http://integrall.bio.ua.pt/>). The incompatible group types of
plasmids were analyzed by using plasma finder database(<https://cge.cbs.dtu.dk/services/PlasmidFinder/>).
the gene information related to antibiotic resistance ~~are-is~~ determined by using the bioinformatics database
card (the comprehensive antimicrobial resistance database (<https://card.mcmaster.ca/home>)). Further,
seven conserved housekeeping genes (*rpoB*, *gapA*, *mdh*, *pgi*, *phoE*, *infB* and *tonB*) are analyzed for
MLST typing by using *Klebsiella pneumoniae* MLST typing database of BIGSDB
Pasteur(<https://bigsdb.pasteur.fr/>). The species of strains are identified by using the ribosomal MLST
database (rMLST, <https://pubmlst.org/species-id>).

**Structural analysis of multi-drug resistance plasmids**

According to the annotation results of ~~the~~ card database, ~~multi-multi~~-drug resistant plasmids pm911-1.1 and
ps90-2.3 were selected, and the plasmid map was drawn by software SNAPGENE software (from
Insightful Science; available at snapgene.com). NCBI was used to blast plasmid drug resistance gene
fragments, and EasyFig (Sullivan et al.,2011) was used to compare the differences of drug resistance
genes and their upstream and downstream drug resistance ~~gene-gene~~-related elements (similarity $\geq 95\%$).
Through plasmid database PLSDB(<https://ccb-microbe.cs.uni-saarland.de/plsdb/>) Search for similar
plasmids (the search conditions are limited to max.p-value = 0, max.distance = 0.04, per. Ident $\geq 60\%$),
and collect their isolation year, country and host information. Some similar plasmids (similarity $\geq 90\%$)
were selected for plasmid genome difference comparison, which was realized by software mauve

(Darling et al.,2004).

**Construction of Bacterial Genome-Wide Phylogenetic Tree**

According to the results of similar plasmids retrieved from PLSDB database (similarity \geq 95%),
~~*Klebsiella-K. pneumoniae*~~ isolates of some plasmids were selected for MLST typing, the
bacterial
genome-wide phylogenetic tree was constructed by using the tool REALPHY 1.13 (Bertels et al.,2014),
and the ST type, host, country, isolation date and other information of the bacterial isolates were
collected for epidemic distribution analysis.

**Results**

*Antimicrobial Phenotype of ~~Klebsiella-K. Pneumoniae~~ strains*

Overall, the antimicrobial phenotype of the four ~~*Klebsiella-K. pneumoniae*~~ strains was different.
~~*Klebsiella-K. pneumoniae*~~ strain S90-2 showed ~~a~~-multiple resistance to 9 first-line antibiotics,
including
ampicillin, cefuroxime, cefazolin, ceftriaxone, cefepime, gentamicin, chloramphenicol, levofloxacin and
~~sulfaxazole/sulfisoxazole~~/trimethoprim. The ~~*Klebsiella-K. pneumoniae*~~ strain S141 was highly resistant
to ampicillin,
cefuroxime, cefazolin and gentamicin. The ~~*Klebsiella-K. pneumoniae*~~ M911-1 was only resistant
to
ampicillin, however ~~*Klebsiella-K. pneumoniae*~~ strain S130-1 was susceptible to 14 drugs (Table 1). All the 4
~~*Klebsiella-K. pneumoniae*~~ strains were susceptible to piperacillin/tazobactam, carbapenem
meropenem,
amikacin and tigecycline (Table 1).

*Genomic Structure and Composition of ~~Klebsiella-pneumoniae~~ strains*

The full length of chromosome of ~~*Klebsiella-K. pneumoniae*~~ S90-2 is 5,374,786kb, belonging to ST629.
The chromosome genome carries 12 gene islands and 30 antimicrobial resistance genes, including
~~*bla*_{CTX-M-14}~~, ~~*fosA6*~~, ~~*aac(3)-Iid*~~ and ~~*bla*_{SHV-11}~~. It carried three plasmids, named ~~as~~-pS90-2.1 (110,388bp,
IncFIB(pKPHS1), pS90-2.2 (109,675bp, IncFIA(HI1)/ IncFII(K)) and pS90-2.3 (57,825 bp, IncR).
pS90-2.3 carried 9 antimicrobial resistance genes, including ~~*mphA*~~, ~~*dfrA12*~~, ~~*aadA2*~~, ~~*qacEdelta1*~~, ~~*sul1*~~,
~~*tet(A)*~~, ~~*aph(3')-Ia*~~, ~~*sul2*~~ and ~~*aph(3'')-Ib*~~, however pS90-2.1 and pS90-2.2 did not carry any antimicrobial
resistance genes. The three plasmids contain 1-2 gene islands, and pS90-2.2 and pS90-2.3 also contain 1
prophage, respectively (Table 2, ~~supplementary-Supplementary~~ Table S1).

The full length of chromosome S141 is 5,383,698 bp, belonging to ST1662. The genome carries 9
gene islands and 27 antimicrobial resistance genes, including ~~*fosA5*~~ and ~~*bla*_{SHV-217}~~. S141 carried two

plasmids named as pS141.1 (194,302 bp, IncFIB (K(pCAV1099-114)/repB) and ps141.2 (112,160bp,
IncFIB (pKPHS1)). pS141.1 contains five gene islands_s, and pS141.2 has a gene island. In addition, only

plasmid pS141.1 carries a drug-resistant efflux pump gene *adeF* (Table 2, ~~supplementary-Supplementary Table S1~~). The
full length of ~~the~~ chromosome of isolate M911-1 is 5,211,192bp. It may be a new ST type that not retrieved
in MLST database. The genome carries 15 gene islands and 27 drug resistance genes, including mainly
*bla*_{SHV-1} and *fosA6*. M911-1 carried three plasmids, which were named ~~as~~-pM911-1.1 (75,711bp, novel),
pM911-1.2 (85,824 bp, IncR/IncFII(pCTU2)) and pM911-1.3 (21,377bp, novel). Plasmid pM911-1.1
carried three drug resistance genes *qnrS1*, *bla*_{LAP-2} and *tet(A)*, and the other two plasmids did not carry
drug resistance genes. pM911-1.1 contains two gene islands, pM911-1.2 has one gene island and one
prophage, and pM911-1.3 has one ~~prephage-prophage~~ (Table 2, ~~supplementary-Supplementary Table S1~~). The full
length of
205 chromosome S130-1 is 5,249,027 bp, belonging to ST3753. The genome carries 10 gene islands and 27
drug resistance genes, including *bla*_{SHV-11} and *fosA6*. S130-1 carries a plasmid named pS130-1
(150,355bp, IncFIB(K)), which carries only one drug resistance gene *tet(A)* and three gene Islands (Table
2, ~~supplementary-Supplementary Table S1~~).

**Homology analysis of multi-drug resistance plasmids pM911-1.1 and pS90-2.3**

Plasmid pM911-1.1 has a full length of 75711bp. There are conjugate transfer regulatory genes
including *TrbB*, *TraY*, *TraL*, *TraI*, and *TraJ* on gene island pM911-1.1-GI-1 (3- 35,601bp), but *finO* gene
and complete *tra/trb* gene cluster are not found. Therefore, it does not have the ability of conjugate
transfer. There was also an *IntI2* transposase gene on the gene island pM911-1.1-GI-1 (40,687 -
45,449bp), but no ~~any antimicrobial gene was found on~~ ~~antimicrobial gene was found~~ upstream and downstream of the
IntI2 gene. In

the PlasmaFinder database, the same incompatible group type as the replication regulatory protein gene
*repA* of the plasmid was not found, so we speculate that the plasmid belongs to a novel incompatible
group (Figure 1A).

Further analysis showed that pM911-1.1 carried an MDR region (size 13,104bp, 50,659 – 63,762bp,
Figure 2A) containing ~~ings~~ three drug resistance genes *qnrS1*, *bla*_{LAP-2} and *tet(A)* and their upstream and
downstream transposable elements. This region was highly homologous with *Klebsiella-K. pneumoniae*
plasmid pCRKP78R-4-tetA and *Klebsiella-K. pneumoniae* plasmid p4_L39 in Zhejiang of China, *Raoultella*
*ornithinolytica* plasmids pWP8-W19-CRE-01_3 in Tokyo of Japan and *Klebsiella grimontii* plasmid
p2481359-2 in Swiss (Figure 2B, ~~supplementary-Supplementary Table S2~~). In addition, a length 1,180bp gene fragment
(54,463-55,643bp) in pM911-1.1 MDR region contains *iseC1*, *ISEc1*, *insC21*, *ISRso10* Transposase and
*ISMaq2* *insD*, while the corresponding region is complete IS3 family transposase genes in homologous
plasmids pCRKP78R-4-tetA, p4_L39, pWP8-W19-CRE-013 and p2481359-2. In contrast, the
tetracycline resistance gene *tet(A)* (60,925–62,124bp) carried by plasmid pM911-1.1 was missing in the
corresponding fragment of plasmid p2481359-2 (Figure 2B).

Plasmid pS90-2.3, a total length of 57,825bp, belongs to IncR incompatible group. The gene island
pS90-2.3-GI-2 (47,134-52,200bp) mainly contains TnAs1, IS15DIV and IS26 family transposase genes
(Figure 1B). In addition, 63.40% of the region of the plasmid is composed of a prophage structure
(19,709- 56,370bp). Except for the phosphotransferase gene *aph(3'')-Ib* (56,431-57,234bp) which
mediates aminoglycoside drug tolerance, other antimicrobial resistance genes, and an IntI1 integron
structure are located in the prophage (Figure 1B). According to the antimicrobial resistance gene carried
by pS90-2.3, it is further divided into two ~~MDR-MDR~~, including MDR1 region (Figure 3A, size
13,626bp,26,378-40,003bp) carrying *mphA*, *dfrA12*, *aadA2*, *qacEdelta1* and *sul1* and MDR2 region
(Figure 3C, size 15,907bp, 41,417-57,234bp) carrying *tet (A)*, *aph (3') - ia*, *sul2* and *aph (3' ' - ib)*.
According to gene traceability analysis, a gene fragment (size 7,644 bp, 32,359- 40,003 bp) in MDR1
region is highly homologous with partial fragments of *Escherichia-E. fergusonii* plasmid pEF01 in Zhejiang
of China, *Klebsiella-K. pneumoniae* plasmid p2018c01-046-3 in Taiwan-of China, *Klebsiella-K. pneumoniae*
plasmid pMV-u1-SK2-O-a in Switzerland and ~~Salmonella-S. enterica serovar Typhimurium~~ plasmid p24362-1 in the
United States. This structure is also highly homologous to some chromosomal genomes of *Citrobacter freundii*
Cf.1 isolated from Guangxi, China (Figure 3B, Supplementary Table S2). Compared with pS90-2.3
MDR1 structure, plasmids pMV-u1-SK2-O-a and p2018C01-046-3 lacked *IS26 tnp* fragment (39,299-
40,003bp), but *Citrobacter freundii* Cf.1 and plasmid p24362-1 lacked *TnAs3 tnpM* fragment (32,359
247 -32,709bp) and *IS26 tnp* fragment (39,299-40,003bp) (Fig. 3B). For pS90-2.3 MDR2, no fragments
highly homologous to this region were found (Figure 3C).

**Prevalence and distribution of multi-drug resistant plasmids**

In the PLSDB database, we screened a total of 155 similar plasmids. From 2016 to 2021, there are
61 plasmids highly similar to pM911-1.1. 32 of them (52.46%) were found in 2020 and 38 plasmids
(62.30%) mainly from China. In addition, 68.85% (42 / 61) came from human host bacteria, and the
other came from host bacteria of pigs and unknown origin (Figure 4A). 94 plasmids were highly similar
to pS90-2.3. From 2014 to 2021, they were also mainly isolated in 2020 (42.43%-,38 / 94) and 17.02%
(16 / 94) from China. 36.17% (34 / 94) from human host bacteria, and the other from host bacteria of
dogs, cats, pigs, ducks, egrets, and ~~o f~~ unknown origin (Figure 4B). pM911-1.1 and pS90-2.3 are
homologous with some plasmids in the database (similarity \geq 90%) in partial gene structure regions.
Although the composition is different, their main framework structure remains unchanged. These
plasmids also take ~~Klebsiella-K. pneumoniae~~ and ~~Escherichia-E. coli~~ as the main bacteria hosts
(Supplementary Figure S1).

Formatted: Font: Italic

Formatted: Font: Not Italic

**Phylogenetic analysis based on the whole genome of *Klebsiella-K. pneumoniae* carrying similar plasmids**

Among 155 similar plasmids, we screened 13 *Klebsiella-K. pneumoniae* host bacteria carry similar
plasmid (similarity \geq 95%) with pM911-1.1 and pS90-2.3, and *Klebsiella-K. pneumoniae* isolates S210-3,
BS433-2, BM334-2 (isolated from human) and M63-1 (isolated from panda) isolated in our laboratory to
construct phylogenetic tree base the whole bacteria genome. Phylogenetic analysis showed that 10
*Klebsiella-K. pneumoniae* isolates belonged to ST11 type and located in the same branch. Among them, the
six strains were highly homologous with *Klebsiella-K. pneumoniae* M911-1, and the four strains were
highly homologous with *Klebsiella-K. pneumoniae* S90-2. *Klebsiella-K. pneumoniae* 19KM1053 was isolated
from domestic cats and *Klebsiella-K. pneumoniae* KP18-29 is unknown, and the other strains are from
human samples (Figure 5, red branch). In addition, the *Klebsiella-K. pneumoniae* 2e (ST4024) carrying
plasmid homologous to pS90-2.3 is in the same branch as *Klebsiella-K. pneumoniae* BM334-2 (ST86) from
human samples, the *Klebsiella-K. pneumoniae* 2018C01-046 (novel ST type) is located in the same branch
as M911-1 (novel ST type), and the SCKP020135 (ST1 type) is in the same branch as M63-1 (ST628
type, isolated from panda) (Figure 5). These *Klebsiella-K. pneumoniae* isolates were mainly isolated from
human samples in China from 2010 to 2019 (Figure 5).

**Discussion**

In this study, we evaluated the antimicrobial resistance phenotype of four *K. pneumoniae* isolates
from wild Chukar Partridge, Red-breasted Parakeet, Sun Parakeet and Black-collared Starling (Table 2).
Studies have shown that wild birds can carry extended spectrum β -Lactamases (ESBL) mediated by
*bla*_{CTX-M} and *bla*_{SHV} from mainly highly pathogenic multidrug-resistant *Escherichia coli* (Ashton et al.,
2015; Sullivan et al.,2011; Darling et al.,2004), *K. pneumoniae* (Darling et al.,2014), *Pseudomonas spp.*
(*Borges et al.,2017*) and *Campylobacter jejuni* (Carroll et al.,2015). Therefore, wild birds have become
an important reservoir host for antimicrobial-resistant bacteria.

The genome of all the *K. pneumoniae* isolates in this study carried the SHV resistance gene (Table
2). This gene SHV type shares about 68% homology with TEM type based on the amino acid sequence
and their frame structure was similar, mainly in *Escherichia-E. coli*, *K. pneumoniae* and *Pseudomonas*
*aeruginosa* (Rybak et al.,2022; Oteo et al.,2018). *bla*_{SHV-1} carried by S90-2 and M911-1 is the earliest
described hydrolase gene in SHV ESBL family and was carried by plasmids or chromosomes of most *K.*
*pneumoniae* and less *Escherichia-E. coli* (GC Rodrigues et al.,2021). The *bla*_{SHV-11} ESBL carried by *K.*
*pneumoniae* S130-1 was first found in the *K. pneumoniae* isolate from Switzerland (Aksomaitiene et
al.,2019), mediating β -lactam drugs tolerance and involving with-nucleotide excision repair, mismatch
repair, DNA replication, aromatic compound degradation, nitrogen metabolism and amino acid

metabolism regulation. *bla*_{SHV-11} also induces DNA damage repair in coordination with *dnaJ, ligA, mutS*,
*recA* and *recF*, which is conducive to maintaining the integrity of the genome (Bradford et al.,2001).
*bla*_{SHV-217} carried by S141 was found late and there is little relevant information. All the gene sequences
of *bla*_{SHV-217} are from *K. pneumoniae*. Therefore, **it** can mediate the tolerance of penicillin and
cephalosporins, which is consistent with the Resistance phenotype of ampicillin in this study (Table 1).

The chromosomes of S90-2, M911-1₂ and S130-1 carry **the** resistance gene *fosA6* which ~~is~~ mediates
high-level fosfomycin resistance and was first found in the clinical strain *Escherichia-E. coli* producing
*bla*_{CTX-M-2} enzyme in a hospital in the United States. Generally, the *FosA6* is highly homologous to the
genomic fragments of many strains of *K. pneumoniae* (Lee et al.,2012). Another homologous gene *fosA5*
carried by S141 was found in the ESBL isolate *Escherichia-E. coli* E265, which also mediated high-level
fosomycin resistance. Gene tracing analysis shows that *fosA5* in plasmid may come from *K. pneumoniae*
CG4 and mediated by the insertion sequence IS10 (Bradford et al.,1999). In our study, it was also found
that the chromosome of S90-2 carried ESBL gene *bla*_{CTX-M-14} and acetyltransferase gene *aac(3)-IId*,
which mediated β - lactams and aminoglycosides resistance (Table 1). Therefore, there may be **a** high level
**of** frequent gene exchange among Enterobacteriaceae strains.

The lateral transfer of resistance genes is usually mediated by mobile genetic elements (such as
insertion sequences, transposons, integrons and prophages, etc.) (Van Boeckel et al.,2017). The MDR
fragments carried by multi-drug resistance plasmid pM911-1.1 and pS90-2.3 are highly homologous with
the fragments of chromosomes or plasmids from many *Enterobacteriaceae* strains (Figure 2 and Figure 3,
Table 2), **indicating-indicates** that there may be a variety of mechanisms for resistance gene transfer. The MDR
region of pM911-1.1 is composed of *tnpR-qnrS1-insC21-transposase-insD-bla*_{LAP-2}-*tnpA-tet(A)-tnp*.
There are transposase genes in the upstream and downstream of the resistance genes *qnrS1* and *bla*_{LAP-2}.
The two resistance genes are ~~obviously~~ separated by transposase genes ISEc11 *insC21*, ISRso10
Transposase and ISMaq2 *insD*, and no longer share Tn3 transposase genes (Figure 2A). It is worth noting
that there is only one complete IS3 family transposase gene between the genes *qnrS1* and *bla*_{LAP-2} in the
homologous reference sequence (Figure 2B). Therefore, it is speculated that the MDR of pM911-1.1 can
be transferred horizontally mediated by ISEc11, ISRso10 and ISMaq2 transposase, while the resistance
gene cluster in homologous reference plasmid (plasmids pCRKP78R-4-tetA, p4_L39,
pWP8-W19-CRE-01_3 and p2481359-2, Figure 2) only occurred a separate horizontal transfer of *qnrS1*
or *bla*_{LAP-2}. The resistance gene *qnrS1* in MDR fragment of pM911-1.1 was first found in conjugated
transfer plasmid from *Shigella flexneri*, which mediates low-level fluoroquinolones resistance
(Nüesch-Inderbinen et al.,1997), and *bla*_{LAP-2} is an β - Lactamase resistance gene from *Enterobacter*
*cloacae* (Miriyala et al.,2020). *tet(A)*, tetracycline efflux pump gene, was first found in many

Gram-negative bacteria (Guo et al.,2016), but plasmid p2481359-2 lacked *tet(A)* (Figure 2). Therefore,
the MDR region may have evolved by multiple transposable or homologous recombination events.

According to the PlasmidFinder database, pM911-1.1 may be a new type of plasmid incompatible
group, and the homology between the *rep* gene of pM911-1.1 and the *rep* gene of IncFII (pCRY)
incompatible plasmid was only 81.32%. IncFII plasmids are widely popular all over the world, and have
become important vectors of ESBLs such as *bla*_{NDM}, *bla*_{SHV}, *bla*_{OXA} and *bla*_{KPC}, which mediate high
levels of β -lactam resistance, including *K. pneumoniae*, *Escherichia-E. coli* and *Enterobacteriaceae*
*cloacae* (Ma et al.,2015; Hata et al.,2005; Huang et al.,2008; Aldema et al.,1996). Plasmid pCRY was
first found in *Yersinia pestis* isolate 91001(Wu et al.,2019), but another study confirmed that the
similarity of gene *repA* between the ~~multi-drug-multidrug~~-resistant plasmid pMET1 carried by *K. pneumoniae*
clinical isolate and the plasmid pCRY was $\geq 95\%$ (Bourouis et al.,2015), indicating that the plasmid
carried by the bacterial host may undergo natural evolution in the process of adapting to the environment.

The MDR1 fragment carried by plasmid ps90-2.3 is composed of IS26 transposase-*mphA*-IS6100
*tmp*-IS26 *tmp*-TnAs3 *tmpM-intI1-dfrA12-aadA2-qacEdelta1-sul1*-IS26 *tmp* (Figure 3A), which mediates
the resistance of macrolides, trimethoprim, aminoglycosides, and sulfonamides (Simner et al.,2018;
Doumith et al.,2017; Song et al.,2004; Soler et al.,2008; Pawlowski et al.,2018). Its core structure is
TnAs3 *tmpM-intI1-dfrA12-aadA2-qacEdelta1-sul1*-IS26 *tmp*, in which exists an integron structure with
the size of 4,199 bp (*intI1-dfrA12-aadA2-qacEdelta1-sul1*). It is reported that the classical structure of
the integron mainly exists in the chromosome of *Escherichia-E. coli* and *K. pneumoniae* (similarity 100%),
however, Its upstream and downstream ~~are~~ lack of transposase genes (Figure 3B). *tmpM* and *tmp* genes
locates in upstream and downstream of the integron structure in pS90-2.3, thus this fragment constitutes
both the integron structure and the transposon structure. Therefore, these results suggest that the
transposase gene promotes the transfer of integron fragments to plasmids and further enhances the
transmission of drug resistance genes (Thungapathra et al.,2002; Chen et al.,2007).

The MDR2 region of pS90-2.3 contains four drug resistance genes, which mainly mediate the
tolerance of tetracyclines, aminoglycosides and sulfonamides. In addition, there are 11 transposase genes
in this region, of which 7 are located on the gene Island (Figure 3C), indicating that MDR2 may be
formed by ~~transposon-transposon~~-mediated multiple gene horizontal transfers. It is worth noting that 63.40% of the
pS90-2.3 plasmid is a prophage structure (19,709-56,370 bp), which is 100% homologous to the ~~P1-P1~~-like
phage RCS47 (Accession: NC_042128.1) found in *Escherichia-E. coli* 725 (serotype O8:H19). RCS47
carries only one antibiotic resistance gene *bla*_{SHV-2}, and there is multiple transposase genes IS26, IS5 and
IS1 at its upstream and downstream (Kazama et al.,1999). In this study, the plasmid pS90-2.3 carried 9
drug resistance genes, among them the eight drug resistance genes were provided by the prophage

(Figure 1B, Figure 3) except for *aph(3'')-Ib* (56 431-57 234 bp). The same as phage RCS47, this plasmid
also carries multiple transposase genes. These transposases provide conditions for the plasmid to obtain
drug resistance genes and integrate the ~~phage-phage~~-carrying drug resistance genes. Plasmid ps90-2.3 belongs
to IncR, which was first discovered in 2009 and does not have the ability of conjugation and transfer
(Martínez et al.,2007). For the clinical isolates of *K. pneumoniae*, the IncR plasmid mainly carries the
drug-resistant genes of *bla_{KPC-2}*, *bla_{DHA-1}*, *bla_{NDM-1}*, *bla_{VIM-1}*, *qnrS1* or *armA* (Zhao et al.,2015; Petrova et
al.,2011; Billard-Pomares et al.,2014). IncR plasmids can coexist with many types of plasmids such as
IncC, IncN, IncHI and IncFII, and the drug-resistant genes carried by them can be transferred laterally
through transposition or plasmid recombination events, thus promoting the diffusion of drug-resistant
genes among bacterial species (García-Fernández et al.,2009).

There are 155 plasmids highly homologous to pM911-1.1 and pS90-2.3, mainly distributed in China
and its surrounding countries; and some in Europe and the Americas (Figure 4). The homologous plasmid
with pM911-1.1 has been reported in Germany, Thailand, Laos, and China before 2020, it was mainly
reported in China and Japan in 2020; and mainly in China, Thailand, the United States and Switzerland
in 2021. Similarly, the homologous plasmids with ps90-2.3 were ~~mainly-primarily~~ found in the United States,
Canada, Germany, Greece, Egypt, Thailand and China before 2020, and were isolated in Switzerland,
Germany, Chile, Australia, Canada and China in 2020, and it was reported in the Czech Republic, China,
the United States and Switzerland in 2021. Therefore, the above two plasmids closely related to the host
bacteria of humans have been widely spread and distributed in China, and may-be gradually becoming the
dominant plasmid type.

In this study, the habitats of Sun Parakeet and Chukar Partridge carrying drug-resistant bacteria and
migratory birds overlap with human colonies [10], and the migratory behavior of wild_birds may further
increase the risk of the spread of drug-resistant strains and drug-resistant plasmids. The host bacteria of
drug-resistant plasmid are mainly ST11 type *K. pneumoniae* (Figure 5, red branch). In recent ten years,
ST11 type *K. pneumoniae* has become a popular dominant clone in China (Guo et al.,2016). Among
ST11 type *K. pneumoniae*, ESBLs *K. pneumoniae* that widely appear are closely associated with
IncFII-like plasmids (Kocsis et al.,2016). Under the mediation of insertion sequence IS26, a variety of
ESBLs genes can coexist in IncF/IncR type plasmids carried by ST11 type *K. pneumoniae* (Qu et
al.,2019). Plasmids carried by ST11 ESBLs *K. pneumoniae* isolates prevalent in Asia are mainly
recombinant plasmids (Compain et al.,2014). The isolates M911-1 and S90-2 in this study are not ST11
type *K. pneumoniae*, and the host bacteria of homologous plasmid has a close genetic relationship with
other ST type non-drug resistant *K. pneumoniae* (Figure 5). Therefore, ST11 type *K. pneumoniae* may be
an important host of multi_drug resistant plasmids, but multi_drug resistant plasmids can also spread in a

variety of different host bacteria.

**Conclusion**

In ~~a~~ conclusion, *K. pneumoniae* carried by wild birds can carry multi-drug resistance plasmids,
which are closely related to human clinical isolates. These plasmids obtain drug-resistance genes through
a variety of mobile elements and endow the strains with ~~multi-multi~~-drug resistance phenotypes. Therefore,
wild birds may become a potential repository of drug-resistant bacteria with clinically ~~important~~significant
drug-resistant genes, which further increases the possibility of AMRS horizontal transfer between
humans, animals, and the environment, thus constituting a hidden danger to public safety.

**References**

- Aksomaitiene,J., Ramonaite,S., Tamuleviciene,E., Novoslavskij,A., Alter,T., Malakauskas,M. (2019).
Overlap of Antibiotic Resistant *Campylobacter jejuni* MLST Genotypes Isolated From Humans,
Broiler Products, Dairy Cattle and Wild Birds in Lithuania. *Front.microbiol.*10,1377.
- Aldema,M.L., McMurry,L.M., Walmsley,A.R., Levy,S.B. (1996). Purification of the Tn10-specified
tetracycline efflux antiporter TetA in a native state as a polyhistidine fusion protein. *Mol. Microbiol.*
19(1),187-195.
- Ashton, P.M., Nair,S., Dallman,T., Rubino,S., Rabsch,W., Mwaigwisya,S., et al. (2015). MinION
nanopore sequencing identifies the position and structure of a bacterial antibiotic resistance island. *Nat.*
*biotechnol.*33(3),296-300.
- Bertels, F., Silander,O.K., Pachkov,M., Rainey,P.B., van Nimwegen,E.(2014). Automated reconstruction
of whole-genome phylogenies from short-sequence reads. *Mol.biol.evol.*31(5),1077-1088.
- Billard-Pomares,T., Fouteau,S., Jacquet,M.E., Roche,D., Barbe,V., Castellanos,M., et al. (2014).
Characterization of a P1-like bacteriophage carrying an SHV-2 extended-spectrum β -lactamase from
an *Escherichia coli* strain. *Antimicrob. Agents. Ch.*58(11),6550-6557.
- Bourouis, A., Ben Moussa,M., Belhadj,O. (2015).Multidrug-resistant phenotype and isolation of a novel
SHV-beta-Lactamase variant in a clinical isolate of *Enterobacter cloacae*. *J.biomed.sci.* 22(1),27.
- Borges,C.A., Beraldo,L.G., Maluta,R.P., Cardozo,M.V., Barboza,K.B., Guastalli,E.A., et al. (2017).
Multidrug-resistant pathogenic *Escherichia coli* isolated from wild birds in a veterinary hospital. *Avian*
*pathol.* 46(1),76-83.
- Bradford, P.A. (1999). Automated thermal cycling is superior to traditional methods for nucleotide
sequencing of bla(SHV) genes. *Antimicrob. Agents.ch.* 43(12),2960-2963.
- Bradford, P.A. (2001).Extended-spectrum beta-lactamases in the 21st century: characterization,
epidemiology, and detection of this important resistance threat. *Clin.microbiol. rev.* 14(4),933-951.

Burnham,C.D., Leeds,J., Nordmann,P., O'Grady,J., Patel,J.(2017). Diagnosing antimicrobial
resistance. *Nat. rev.Microbiol.*15(11),697-703.

Bush,K., Bradford,P.A.(2020). Epidemiology of β -Lactamase-Producing Pathogens. *Clin.microbiol. rev.*
33(2),e00047-19.

Carroll,D., Wang,J., Fanning,S., McMahon,B.J. (2015).Antimicrobial Resistance in Wildlife:
Implications for Public Health. *Zoonoses.public. hlth.* 62(7),534-542.

Chang,H.H., Cohen,T., Grad,Y.H., Hanage,W.P., O'Brien,T.F., Lipsitch,M. (2015).Origin and
proliferation of multiple-drug resistance in bacterial pathogens. *Microbiol.mol.biol.r.*79(1):101-116.

Chen,Y.T., Lauderdale,T.L., Liao,T.L., Shiau,Y.R., Shu,H.Y., Wu,K.M., et al. (2007).Sequencing and
comparative genomic analysis of pK29, a 269-kilobase conjugative plasmid encoding CMY-8 and
CTX-M-3 beta-lactamases in *Klebsiella pneumoniae*. *Antimicrob.agents.ch.* 51(8),3004-3007.

CLSI. Performance Standards for Antimicrobial Susceptibility Testing. 31st ed.

Compain,F., Frangeul,L., Drieux,L., Verdet,C., Brisse,S., Arlet,G., Decré,D. (2014).Complete nucleotide
sequence of two multidrug-resistant IncR plasmids from *Klebsiella pneumoniae*.- *Antimicrob.*
*Agents.ch.* 58(7),4207-4210.

Cummins,M.L., Sanderson-Smith,M., Newton,P., Carlile,N., Phalen,D.N., Maute,K., et al.
(2020).Whole-Genome Sequence Analysis of an Extensively Drug-Resistant *Salmonella enterica*
Serovar Agona Isolate from an Australian Silver Gull (*Chroicocephalus novaehollandiae*) Reveals the
Acquisition of Multidrug Resistance Plasmids. *mSphere.*5(6),e00743-20.

Darling,A.C., Mau,B., Blattner,F.R., Perna,N.T.(2004). Mauve: multiple alignment of conserved genomic
sequence with rearrangements. *Genome.res.*14(7),1394-1403.

Doumith,M., Findlay,J., Hirani,H., Hopkins,K.L., Livermore,D.M., Dodgson,A., Woodford,N. (2017).
Major role of pKpQIL-like plasmids in the early dissemination of KPC-type carbapenemases in the
UK. *J.antimicrob. chemoth.*72(8),2241-2248.

Eiamphungporn,W., Schaduangrat,N., Malik, A.A., Nantasenamat,C. (2018).Tackling the Antibiotic
Resistance Caused by Class A β -Lactamases through the Use of β -Lactamase Inhibitory Protein.
*Int.j.mol.sci.*19(8),2222.

Fashae,K., Engelmann, I., Monecke, S., Braun, S.D., Ehricht, R.(2021). Molecular characterization of
extended-spectrum β -lactamase producing *Escherichia coli* in wild birds and cattle, Ibadan, Nigeria.
*BMC. vet. Res.*17(1),33.

Guenther, S., Aschenbrenner, K., Stamm, I., Bethe, A., Semmler, T., Stubbe, A., et al. (2012).
Comparable high rates of extended-spectrum-beta-lactamase-producing *Escherichia coli* in birds of
prey from Germany and Mongolia. *PloS. one.*7(12),e53039.

García-Fernández, A., Fortini, D., Veldman, K., Mevius, D., Carattoli, A. (2009). Characterization of
plasmids harbouring *qnrS1*, *qnrB2* and *qnrB19* genes in *Salmonella*.
*J. antimicrob. chemoth.* 63(2),274-281.

GC Rodrigues, J., Nair, H.P., O'Kane, C., Walker, C.A. (2021). Prevalence of multidrug **multi-drug** resistance
in *Pseudomonas* spp. isolated from wild bird feces in an urban aquatic environment.
*Ecol. evol.* 11(20),14303-14311.

Guo, Q., Spychala, C.N., McElheny, C.L., Doi, Y. (2016). Comparative analysis of an IncR plasmid
carrying *armA*, *bla_{DHA-1}* and *qnrB4* from *Klebsiella pneumoniae* ST37 isolates. *J. antimicrob. chemoth.*
(4),882-886.

Guo, Q., Tomich, A.D., McElheny, C.L., Cooper, V.S., Stoesser, N., Wang, M., et al. (2016).
Glutathione-S-transferase FosA6 of *Klebsiella pneumoniae* origin conferring fosfomycin resistance in
ESBL-producing *Escherichia coli*. *J. antimicrob. chemoth.* 71(9),2460-2465.

Hata, M., Suzuki, M., Matsumoto, M., Takahashi, M., Sato, K., Ibe, S., Sakae, K. (2005). Cloning of a
novel gene for quinolone resistance from a transferable plasmid in *Shigella flexneri* 2b.
*Antimicrob. agents. ch.* 49(2),801-803.

Hassell, J.M., Ward, M.J., Muloi, D., Bettridge, J.M., Phan, H., Robinson, T.P., et al. (2019).
Deterministic processes structure bacterial genetic communities across an urban landscape. *Nat.*
*commun.* 10(1),2643.

Huang, Z., Mi, Z., Wang, C. (2008). A novel beta-lactamase gene, LAP-2, produced by an *Enterobacter*
*cloacae* clinical isolate in China. *J. hosp. infect.* 70(1),95-96.

Ji, F., Liu, S., Wang, X., Zhao, J., Zhu, J., Yang, J., et al. (2022). Characteristics of the multiple replicon
plasmid IncX1-X1 in multidrug-resistant *Escherichia coli* from Malayan pangolin (*Manis javanica*).
*Integr. zool.*

Kang, Q., Wang, X., Zhao, J., Liu, Z., Ji, F., Chang, H., et al. (2021). Multidrug-resistant *Proteus*
*mirabilis* isolates carrying *bla_{OXA-1}* and *bla_{NDM-1}* from wildlife in China: increasing public health risk.
*Integr. zool.* 16(6),798-809.

Kazama, H., Hamashima, H., Sasatsu, M., Arai, T. (1999). Characterization of the antiseptic-resistance
gene *qacE* delta 1 isolated from clinical and environmental isolates of *Vibrio parahaemolyticus* and
*Vibrio cholerae* non-O1. *FEMS. Microbiol. Lett.* 174(2),379-384.

Klein, E.Y., Van Boeckel, T.P., Martinez, E.M., Pant, S., Gandra, S., Levin, S.A., et al. (2018). Global
increase and geographic convergence in antibiotic consumption between 2000 and 2015. *PNAS.*
115(15), E3463-E3470.

Kocsis, E., Gužvinec, M., Butić, I., Krešić, S., Crnek, S.Š., Tambić, A., et al. (2016). *bla_{NDM-1}* Carriage

on IncR Plasmid in *Enterobacteriaceae* Strains. *Microb.drug. resist.(Larchmont, N.Y.)*.22(2),123-128.
Laxminarayan, R., Duse, A., Wattal, C., Zaidi, A.K., Wertheim, H.F., Sumpradit, N., et al. (2013).
Antibiotic resistance-the need for global solutions. *Lancet. Infect.dis.* 13(12),1057-1098.
Lee, J.H., Bae, I.K., Lee, S.H. (2012).New definitions of extended-spectrum β -lactamase conferring
worldwide emerging antibiotic resistance. *Med.res.rev.*32(1),216-232.
Loman, N.J., Quick, J., Simpson, J.T. (2015).A complete bacterial genome assembled de novo using only
nanopore sequencing data. *Nat.methods.*12(8),733-735.
501 Ma, Y., Xu, X., Guo, Q., Wang, P., Wang, W., Wang, M. (2015).Characterization of *fosA5*, a new
plasmid-mediated fosfomycin resistance gene in *Escherichia coli*. *Lett.appl.microbiol.* 60(3),259-264.
Marcelino, V.R., Wille, M., Hurt, A.C., González-Acuña, D., Klaassen, M., Schlub, T.E., et al. (2019).
Meta-transcriptomics reveals a diverse antibiotic resistance gene pool in avian microbiomes. *BMC*
*biology*.17(1):31.
Martínez, N., Mendoza, M.C., Rodríguez, I., Soto, S., Bances, M., Rodicio, M.R.(2007). Detailed
structure of integrons and transposons carried by large conjugative plasmids responsible for multi_drug
resistance in diverse genomic types of *Salmonella enterica* serovar Brandenburg.
*J.antimicrob.chemoth.* 60(6),1227-1234.
Miryala, S.K., Anbarasu, A., Ramaiah, S. (2020).Role of SHV-11, a Class A β -Lactamase, Gene in
Multidrug Resistance Among *Klebsiella pneumoniae* Strains and Understanding Its Mechanism by
Gene Network Analysis. *Microb.drug resist.(Larchmont, N.Y.)*.26(8),900-908.
Nüesch-Inderbinen, M.T., Kayser, F.H., Hächler, H. (1997).Survey and molecular genetics of SHV
beta-lactamases in *Enterobacteriaceae* in Switzerland: two novel enzymes, SHV-11 and SHV-12.
*Antimicrob. Agents. Ch.* 41(5),943-949.
Oteo, J., Mencía, A., Bautista, V., Pastor, N., Lara, N., González-González, F., et al. (2018). Colonization
with *Enterobacteriaceae*-Producing ESBLs, AmpCs, and OXA-48 in Wild Avian Species, Spain
2015-2016. *Microb.drug. resist. (Larchmont, N.Y.)*. 24(7),932-938.
Pawlowski, A.C., Stogios, P.J., Koteva, K., Skarina, T., Evdokimova, E., Savchenko, A., Wright, G.D.
(2018). The evolution of substrate discrimination in macrolide antibiotic resistance enzymes.
*Nat.commun.* 9(1),112.
Páll, E., Niculae, M., Brudaşcă, G.F., Ravilov, R.K., Şandru, C.D., Cerbu, C., et al. (2021).Assessment
and Antibiotic Resistance Profiling in *Vibrio* Species Isolated from Wild Birds Captured in Danube
Delta Biosphere Reserve, Romania. *Antibiotics (Basel, Switzerland)*.10(3),333.
Partridge, S.R., Kwong, S.M., Firth, N., Jensen, S.O. (2018).Mobile Genetic Elements Associated with
Antimicrobial Resistance. *Clin.microbiol.rev.*31(4),e00088-17.

Petrova, M., Gorlenko, Z., Mindlin, S. (2011). Tn5045, a novel integron-containing antibiotic and
chromate resistance transposon isolated from a permafrost bacterium. *Res.microbiol.*162(3),337-345.

Qu, D., Shen, Y., Hu, L., Jiang, X., Yin, Z., Gao, B., et al. (2019). Comparative analysis of
KPC-2-encoding chimera plasmids with multi-replicon IncR:IncpA1763-KPC:IncN1 or
IncFIIpHN7A8: IncpA1763-KPC:IncN1. *Infect.drug. resist.*12,285-296.

Ruiz-Ripa, L., Gómez, P., Alonso, C.A., Camacho, M.C., Ramiro, Y., de la Puente, J., et al. (2020).
Frequency and Characterization of Antimicrobial Resistance and Virulence Genes of
Coagulase-Negative *Staphylococci* from Wild Birds in Spain. Detection of tst-Carrying *S.*
*sciuri* Isolates. *Microorganisms.* 8(9),1317.

Rybak, B., Krawczyk, B., Furmanek-Blaszczak, B., Wysocka, M., Fordon, M., Ziolkowski, P., et al. (2022).
Antibiotic resistance, virulence, and phylogenetic analysis of *Escherichia coli* strains isolated from
free-living birds in human habitats. *PLoS one.* 17(1), e0262236.

Simner, P.J., Antar, A.A.R., Hao, S., Gurtowski, J., Tamma, P.D., Rock, C., et al. (2018).Antibiotic
pressure on the acquisition and loss of antibiotic resistance genes in *Klebsiella pneumoniae*.
*J.antimicrob.chemoth.*73(7),1796-1803.

Smith, S., Wang, J., Fanning, S., McMahon, B.J. (2014).Antimicrobial resistant bacteria in wild
mammals and birds: a coincidence or cause for concern? *Irish vet.j.*67(1),8.

Soler Bistué, A.J., Birshan, D., Tomaras, A.P., Dandekar, M., Tran, T., Newmark, J., et al.(2008).
*Klebsiella pneumoniae* multiresistance plasmid pMET1: similarity with the *Yersinia pestis* plasmid
pCRY and integrative conjugative elements. *PLoS one.*3(3):e1800.

Song, Y., Tong, Z., Wang, J., Wang, L., Guo, Z., Han, Y., et al. (2004).Complete genome sequence of
*Yersinia pestis* strain 91001, an isolate avirulent to humans. *DNA.res.*11(3),179-197.

Stokes, H.W., Gillings, M.R. (2011).Gene flow, mobile genetic elements and the recruitment of antibiotic
resistance genes into Gram-negative pathogens. *FEMS microbiol.rev.*35(5):790-819.

Sullivan, M.J., Petty, N.K., Beatson, S.A. (2011).Easyfig: a genome comparison visualizer.
*Bioinformatics (Oxford, England).*27(7):1009-1010.

The European Committee on Antimicrobial Susceptibility Testing. Breakpoint tables for interpretation of
MICs and zone diameters, version 10.0, 2021

Thungapathra, M., Amita Sinha,K.K., Chaudhuri, S.R., Garg, P., Ramamurthy, T., Nair, G.B., Ghosh, A.
(2002).Occurrence of antibiotic resistance gene cassettes *aac(6')-Ib*, *dfrA5*, *dfrA12*, and *ereA2* in class
I integrons in non-O1, non-O139 *Vibrio cholerae* strains in India.
*Antimicrob.agents.ch.*46(9),2948-2955.

Van Boeckel, T.P., Glennon, E.E., Chen, D., Gilbert, M., Robinson, T.P., Grenfell, B.T., et al. (2017).

Reducing antimicrobial use in food animals. *Science*. 357(6358),1350-1352.
Wang, X., Kang, Q., Zhao, J., Liu, Z., Ji, F., Li J, et al. (2020).Characteristics and Epidemiology of
Extended-Spectrum β -Lactamase-Producing Multidrug-Resistant *Klebsiella pneumoniae* From Red
Kangaroo, China. *Front.microbiol*.11,560474.
Wang, J., Ma, Z.B., Zeng, Z.L., Yang, X.W., Huang, Y., Liu, J.H.(2017). Response to Comment on "The
role of wildlife (wild birds) in the global transmission of antimicrobial resistance genes". *Zool. res.*
38(4),212.
Wu, W., Feng, Y., Tang, G., Qiao, F., McNally, A., Zong, Z. (2019).NDM Metallo- β -Lactamases and
Their Bacterial Producers in Health Care Settings. *Clin. Microbiol.rev.* 32(2):e00115-18.
Yuan, Y., Liang, B., Jiang, B.W., Zhu, L.W., Wang, T.C., Li, Y.G., et al. (2021).Migratory wild birds
carrying multidrug-resistant *Escherichia coli* as potential transmitters of antimicrobial resistance in
China. *PloS.one*.16(12), e0261444.
Zhao, J.Y., Mu, X.D., Zhu, Y.Q., Xi, L., Xiao, Z. (2015). Identification of an integron containing the
quinolone resistance gene *qnrA1* in *Shewanella xiamenensis*. *FEMS. Microbiol.lett.* 362(18),fnv146.

Acknowledgments:

Funding:

This study was granted by the Introduction of Leading Talents Program of the Guangdong Academy of Sciences (No. 2016GDASRC-0205) and Open project of Beijing Key Laboratory of captive wildlife technology in Beijing Zoo (ZDK202105).

Author contributions:

The study was designed and supervised by CMW, and XW; JNZ,FJ and MW isolated and identified bacteria isolates; GYD, JHQ, RLZ and XW collected all the samples; XW and BW analyzed the data; JNZ, XW and CMW prepared original draft; XW, JNZ and CMW reviewed and edited the manuscript.

Competing interests:

Authors declare that they have no competing interests.

Data and materials availability:

All accession numbers to bacteria genome or plasmids relating to the paper and deposited in a GenBank database. All data are available in the main text or supplementary materials.

Figure legends

Figure 1. Plasmid profiles of multi-drug resistant plasmids pM911-1.1(A) and pS90-2.3 (B).

Figure 2. Structure of plasmid pM911-1.1 MDR region and its homologous fragment structure. A, Schematic diagram of gene element combination in pM911-1.1 MDR region. **B,** Comparison of pM911-1.1 homologous fragment structure.- red fragment, drug resistance gene; blue fragment, transposase gene. The arrow direction represents the gene coding direction.

Figure 3. Structure of plasmid pS90-2.3 MDR region and its homologous fragment structure. A, schematic representation of gene element assembly in pS90-2.3 MDR1 region. B, comparison of pM911-1.1 homologous fragment structure. C, schematic representation of gene element assembly in pS90-2.3 MDR2 region. red fragment, drug resistance gene; blue fragment, transposase gene; green fragment, integrase gene; purple fragment, ~~prephage~~prophage; cyan, gene island. The arrow direction represents the gene coding direction.

Figure 4. Global distribution of plasmids pM911-1.1, pS90-2.3 homologs. A, Time of isolation, country and host information of pM911-1.1 homologs. B, Time of isolation, country and host information of pS90-2.3 homologs.

**Figure 5. Genome-wide evolutionary tree of plasmids pM911-1.1, pS90-2.3 homologous plasmids**
 ***Klebsiella-K. pneumoniae* host and strains M911-1 and S90-2.3. Red branch, main clustering branch of**
 **similar plasmid hosts. Orange pentagons, pM911-1.1 similar plasmid. Blue diamond, pS90-2.3 similar**

plasmid.

★ pM911-1.1 similar plasmid

◆ pS90-2.3 similar plasmid

N/A:Not Applicable

Table legends

Table 1. Antimicrobial phenotype of *Klebsiella pneumoniae* strains

Name	Host	Non-susceptible phenotype	Susceptible phenotype
S90-2	Alectoris chukar	AMP, CXM, CZO, CRO, FEP, GEN, CHL, LVX, SXT	SAM, TZP, MEM, GEN, AMK, CHL, LVX, SXT, TGC
S141	Psittacula alexandri	AMP, CXM, CZO, GEN	TZP, MEM, GEN, AMK, CHL, LVX, SXT, TGC
M911-1	Aratinga solstitialis	AMP	CXM, CZO, CRO, FEP, SAM, TZP, MEM, GEN, AMK, CHL, LVX, SXT, TGC
S130-1	Sturnus nigricollis	N/A	AMP, CXM, CZO, CRO, FEP, SAM, TZP, MEM, GEN, AMK, CHL, LVX, SXT, TGC

AMP, Ampicillin; CXM, Cefuroxim; CZO, Cefazolin; CRO, Ceftriaxone; FEP, Cefepime; SAM, Ampicillin/Sulbactam; TZP, Piperacillin/Tazobactam; MEM, Meropenem; GEN, Gentamicin; AMK, Amikacin; CHL, Chloramphenicol; LVX, Levofloxacin; SXT, Trimethoprim/Sulfamethoxazole; TGC, Tigecycline; N/A, Not Applicable.

Table 2. Genomic information of ~~*Klebsiella*~~ *K. pneumoniae* isolates from wild birds

Chromosome group or plasmid name of strain	ST Type	PlasmidFinder	Movable resistance determinants
Chr-S90-2	ST629	-	bla _{CTX-M-14} , fosA6 , aac(3)-IId , bla _{SHV-11}
pS90-2.1		IncFIB(pKPHS1)	-
pS90-2.2		IncFIA(HI1)/IncFII(K)	-
pS90-2.3		IncR	mphA , dfrA12 , aadA2 , qacEdelta1 , sul1 , tet(A) , aph(3')-Ia , sul2 , aph(3'')-Ib
Chr-S141	ST1662	-	fosA5 , bla _{SHV-217}
pS141.1		IncFIB(K)(pCAV1099-114)/repB	-
pS141.2		IncFIB(pKPHS1)	-
Chr-M911-1	novel	-	bla _{SHV-1} , fosA6
pM911-1.1		novel	qnrS1 , bla _{LAP-2} , tet(A)
pM911-1.2		IncR/IncFII(pCTU2)	-
pM911-1.3		novel	-
Chr-S130-1	ST3753	-	bla _{SHV-11} , fosA6
pS130-1		IncFIB(K)	tet(A)

Supplementary Figures and tables:

Figure S1. Homologous structure comparison of similar plasmids pM911-1.1, pS90-2.3 based on
software Mauve for multi-drug resistant plasmids.

Table S1. Information of strains

Strain name	Strain accession	Host	Genome size of strain(kb)	Plasmid name	Plasmid accession	Plasmid size of strain(kb)	Number of drug resistance genes	Number of genes	Number of prophage
S90-2	CP06388 1.1	Alectoris chukar	5374.7 86	-	-	-	30	12	0
				pS90-2.1	CP06388	110.38	8	0	1
				pS90-2.2	CP06388	109.67	5	0	1
				pS90-2.3	CP06388	57.825	9	2	1
S141	CP06387 1.1	Psittacula alexandri	5383.6 98	-	-	-	27	9	0
				pS141.1	CP06387	194.30	2	1	5
				pS141.2	CP06387	112.16	0	0	1
M911-1	CP06412 9.1	Aratinga solstitialis	5211.1 92	-	-	-	27	15	0
				pM911-1.1	CP06413	75.711	0.1	3	2
				pM911-1.2	CP06413	85.824	1.1	0	1
				pM911-1.3	CP06413	21.377	2.1	0	0
S130-1	CP06386 5.1	Sturnus nigricollis	5249.0 27	-	-	-	27	10	0
				pS130-1	CP06386	150.35	1	3	0

Table S2. Source of homologous gene fragment in MDR region of drug resistant plasmid

MDR region of drug-resistant plasmid in this study	Name	Type	Accession	Strain	Host	Date	Country
	pCRKP78R-4tetA	plasmid	CP0662 57.1	Klebsiella pneumoniae strain CRKP78R	Homo sapiens	2018.0 5.23	China:Hangzhou
	p4_L39	plasmid	CP0339 57.1	Klebsiella pneumoniae strain L39_2	Homo sapiens	2018	China:Zhejiang
pM911-1_MDR	pWP8-W19-CR E-01_3	plasmid	AP0222 71.1	Raoultella ornithinolytica strain WP8-W19-CR E-01	wastewater treatment plant effluent	2019.0 2.05	Japan:Tokyo
	p2481359-2	plasmid	CP0673 82.1	Klebsiella grimontii strain 2481359	Homo sapiens	2015.1 0	Switzerland
pS90-2.3_MDR1	pEF01	plasmid	CP0408 06.1	Escherichia fergusonii strain EFCF056	chicken	2017.0 5.04	China:Zhejiang
	pMV-u1-SK2-O-a	plasmid	CP0858 67.1	Klebsiella pneumoniae strain	environmental swab	2020	Switzerland

p2018C01-046-3	plasmid	CP0443 71.1	MV-u1-SK2-O Klebsiella pneumoniae strain 2018C01-046	veterinary clinic Homo sapiens	2018	China: Taiwan
Cf.1	chromosome	CP0856 42.1	Citrobacter freundii strain Cf.1	bullfrog	2020.1 0.11	China: Guangxi Zhuang Nationality Autonomous Region
p24362-1	plasmid	CP0513 79.1	Salmonella enterica subsp. enterica serovar Typhimurium strain CVM 24362	swine	2002	USA:MO

730
731
732
733

. ARGs . IS/Tn . integron ➔ *rep.sop,par*
➔ *tra/trb* Transfer system — Genomic island — Prophage

(A)

(B)

A

B

* pM911-1.1 similar plasmid

◆ pS90-2.3 similar plasmid

N/A: Not Applicable

Response to Reviewer Comments

Dear editor and reviewers,

Manuscript ID: Spectrum02691-22 entitled “Genomic characteristics and molecular epidemiology of multidrug resistant *Klebsiella pneumoniae* carried by wild birds” which we submitted to *Microbiology Spectrum* has been reviewed. Thank the reviewers for their valuable comments. Next, we will answer and explain the questions one by one.

Reviewer #1 (Comments for the Author):

The study investigate the genomic features of pathogen bacteria from the wild birds. I have some major concerns:

(1) the title is so large. This study only focus on several species, and 'wild birds' in the title is over-stated.

Responses:

We collected 125 samples from 50 wild bird species including chuckar, red-breasted parakeet, sun parakeet, black-collared starring, japanese white-eye, chestnut-tailed minla, asian barbed owlet, great egret, red-contaminated bulbul, collared scops owl and so on between 2018-2022. 356 pathogenic bacteria of various conditions were isolated from these bird samples which belong to *Enterobacter*, *Aeromonas*, *Proteus*, *Enterococcus* and *Staphylococcus*, among others. Therefore, this study detected chuckar, red-breasted parakeet, sun parakeet, black-collared starring carried *K. pneumoniae* isolates which carried multidrug resistance genes and mobile genetic elements in a large-scale routine screening rather than a few bird species.

(2) what's reason for choose these species?

Responses:

As stated above, we detected the isolates from chuckar were resistant to 9 drugs, the isolates from the red-broken parakeet were resistant to 3 drugs, the isolates from the sun parakeet were resistant to 1 drug, and the isolates from black-colored starring were all sensitive to the drugs used in this study in a routine screen. To assess the association between drug resistance genes and phenotypes, we sequenced the whole genome of the above four *K. pneumoniae* strains and found multiple drug resistance genes and mobile elements on their genomes. It is worth noting that the black-collared starring isolates, although not showing a resistance phenotype, it also carries drug-resistant genes such as β -lactams and tetracyclines.

(3) the samples size is small, and the author only study one bacteria. Thus, the novelty and importance is low.

Responses:

As a common clinical condition pathogen, tens of thousands of human deaths each year are directly related to *K. pneumoniae* infection, which is particularly serious in ICU wards^[1-2]. *K. pneumoniae* is the main carrier of NDM, IPM, and KPC β -lactamase, and widely mediates the spread and transfer of the above resistance genes. At the same time, the bacteria often carry resistance genes of aminoglycosides, tetracyclines, fluoroquinolones and other drugs^[3-5]. Therefore, this strain is the focus of drug resistance research. The mechanism of *Klebsiella* carrying drug resistance genes is relatively complex, which also provides conditions for its resistance to multiple drugs. In our previous research, it was observed that *Klebsiella* carrying multiple replicon plasmids has stronger drug resistance and lower adaptive cost^[6]. In this study, the drug resistance of *K. pneumoniae* carried by wild birds is mainly mediated by plasmids. By querying the NCBI PubMed database, no similar structure and drug resistance gene combination of the plasmid resistant region in this study have been found in wild birds, but this structure has high homology with the drug resistance region carried by human isolated strains. It is difficult to obtain wild animal samples, and most isolates also show drug resistance sensitive phenotype, while the strains involved in this study show strong drug resistance, which should be paid attention to. This study provides new insights into the emergence and prevalence of drug resistance.

References:

- [1] Paterson DL, Ko WC, Von Gottberg A, et al. International prospective study of *Klebsiella pneumoniae* bacteremia: implications of extended-spectrum beta-lactamase production in nosocomial Infections. *Ann Intern Med.* 2004;140(1):26-32.
- [2] Gu D, Dong N, Zheng Z, et al. A fatal outbreak of ST11 carbapenem-resistant hypervirulent *Klebsiella pneumoniae* in a Chinese hospital: a molecular epidemiological study. *Lancet Infect Dis.* 2018;18(1):37-46.
- [3] Wu W, Feng Y, Tang G, et al. NDM Metallo- β -Lactamases and Their Bacterial Producers in Health Care Settings. *Clin Microbiol Rev.* 2019;32(2):e00115-18.
- [4] Wyres KL, Lam MMC, Holt KE. Population genomics of *Klebsiella pneumoniae*. *Nat Rev Microbiol.* 2020;18(6):344-359.
- [5] Grundmann H, Glasner C, Albiger B, et al. Occurrence of carbapenemase-producing *Klebsiella pneumoniae* and *Escherichia coli* in the European survey of carbapenemase-producing *Enterobacteriaceae* (EuSCAPE): a prospective, multinational study. *Lancet Infect Dis.* 2017;17(2):153-163.
- [6] Wang X, Zhao J, Ji F, et al. Multiple-Replicon Resistance Plasmids of *Klebsiella* Mediate Extensive Dissemination of Antimicrobial Genes. *Front Microbiol.*

2021;12:754931.

(4) as I know, now more and more study on the wild birds focus on the virus, which may have the negative effects on the human health. Thus, I don't think this study will bring the interesting for the readers.

Responses:

Wild birds have a high degree of overlap with human activity areas. Wild birds such as sparrows, eurasian blackbird, parrots, egrets and turtledoves are common birds in urban and rural areas, which are significantly affected by human activities. It is well known that the overuse and discharge of antibiotics by humans significantly affect the environmental flora and lead to the evolution of high-level bacterial resistance. The study by Hassell JM, et al. pointed out that non-random processes structure bacterial genetic communities in urban wildlife and demonstrate that communities of avian-borne bacterial genes are shaped by the assemblage of co-existing avian, livestock and human communities, and the habitat within which they exist. The study also proposes a process of drug resistance gene transfer from humans → domestic animals → feces → birds ^[1]. In addition, the migration behavior of birds makes it difficult to control the spread of drug-resistant genes and drug-resistant strains, which poses a serious threat to human health and public health security. For a long time, people may have ignored the important role of wild animals in the epidemic of drug resistance. The important role of wild animals in the epidemic of drug resistance may have been overlooked for a long time. Therefore, it is urgent and important to pay attention to and study the phenomenon of drug resistance transfer caused by wild birds.

References:

[1] Hassell JM, Ward MJ, Muloi D, et al. Deterministic processes structure bacterial genetic communities across an urban landscape. *Nat Commun.* 2019;10(1):2643.

(5) the authors explore the genomic features on this bacteria, and this study may be fit some microbial journal on genomic studies.

Responses:

Our study sequenced the whole genome based on phenotype research to explain the important role of wild birds in the spread of microbial drug resistance by bioinformatics and comparative genomics methods. Genome sequencing and bioinformatics analysis are only a technical means, not the purpose of this study. Identifying the characteristics of genome which is the key link for understanding the origin and evolutionary recombination process of gene fragments mediating drug

resistance. In the field of microbiology, since drug resistance studies are often limited to humans and domestic animals, and wild bird samples are not easily available, the analysis of wild birds accounts for a small proportion of drug-resistant bacteria research. We believe that it is in line with the Microbiology Spectrum magazine's area of interest. Therefore, the key point of this study is to isolate plasmid mediated multidrug resistance *K. pneumoniae* from wild bird samples. Without supervision, the spread of drug resistance mediated by wild birds will cause incalculable losses and risks.

Reviewer #2 (Comments for the Author):

Manuscript ID # Spectrum02691-22 entitled " Genomic characteristics and molecular epidemiology of multidrug resistant *Klebsiella pneumoniae* carried by wild birds" by Xue Wang et al. reports MDR *Klebsiella pneumoniae* were isolated from fresh feces of captured wild bird and showed drug-resistant phenotypes conferred mainly by multidrug-resistant plasmids.

The major comments:

The problem of antibiotic resistance (AMR) caused by antibiotic drug abuse is becoming more and more serious. At present, it has become an important challenge of global public health. The multidrug-resistant *Klebsiella pneumoniae* isolates carried by wild birds had drug-resistant phenotypes conferred mainly by multidrug-resistant plasmids. Wild bird habitats overlap to a greater extent with human and livestock habitats, and further increase the potential for horizontal transfer of multi-drug resistant bacteria between human-animal-environment. This study is very importance to explore the relationship between wild birds and the transmission of multi-drug resistant strain.

Some minor comments:

Line 136, "DNA;" revised to "DNA."

Line 138, ";" revised to "and"

Line 185, "5,374,786kb" should be "5,374,786 b"

Line 211, "5711bp" should be "5,711bp"

Table 1 and Table S1, bird host name should be revised to "English name"

I recommend acceptance after the above minor modifications.

Responses:

Thanks to the reviewer for the suggestion, the above question will be revised.

Line 132, "DNA;" revised to "DNA."

Line 134, ";" revised to "and".

Line 178, "5,374,786kb" revised to "5,374,786bp".

Line 203, "75711bp" revised to "75,711bp".

The host name in Table 1 and Table S1 has been changed to english name. For

details, please refer to the attachments “Table 1” and “Table S1”.

Reviewer #6 (Comments for the Author):

Interesting/relevant; but suggest to enlarge the number and species of animals

Responses:

This study is a long-term survey. From 2018 to 2022, we collected 125 wild bird samples, involving a total of 50 species of birds, such as chuckar, red-breasted parakeet, sun parakeet, black-collared starrng, japanese white-eye, etc. 356 strains of various opportunistic pathogens such as *Enterobacter*, *Aeromonas*, *Proteus*, *Enterococcus* and *Staphylococcus* were isolated from these bird samples. Among the 4 isolates involved in this study, 2 were multidrug-resistant bacteria and 2 were sensitive to the drugs we used. In order to compare the genomic differences between multidrug-resistant bacteria and sensitive bacteria, 4 strains of *K. pneumoniae* closely related to human clinic were selected for in-depth research, with a view to answering the important role played by wild birds in the spread of bacterial drug resistance.

Reviewer #7 (Comments for the Author):

This is an interesting manuscript that may be interesting to those who are working in this field. There are multiple typos that need to be fixed. Please see the enclosed marked manuscript.

Responses:

Thanks to the reviewer for the suggestion, the above question will be revised.

Line 22, “*Klebsiella pneumoniae*” revised to “*Klebsiella pneumoniae*”.

Line 24, “was” revised to “were”.

Line 25, “ST629” revised to “ST629,”.

Line 26, “genes” revised to “genes,”; “*bla*_{CTX-M-14} and *bla*_{SHV-11}” revised to “*bla*_{CTX-M-14} and *bla*_{SHV-11}”.

Line 27, “*qacEdelta1*, *sul1* and *aph(3')-Ib* etc” revised to “*qacEdelta1*, *sul1* and *aph(3')-Ib* etc”.

Line 28, “*bla*_{SHV-217}” “*bla*_{SHV-1}” “*fosA6*” revised to “*bla*_{SHV-217}” “*bla*_{SHV-1}” “*fosA6*”.

Line 29, “*qnrS1*” “*bla*_{LAP-2}” “*tet(A)*” “*bla*_{SHV-11}” “*fosA6*” revised to “*qnrS1*” “*bla*_{LAP-2}” “*tet(A)*” “*bla*_{SHV-11}” “*fosA6*”.

Line 30, “*tet(A)*” revised to “*tet(A)*”.

Line 35, “*Klebsiella pneumoniae*” revised to “*K. pneumoniae*”.

Line 36, “*Klebsiella pneumoniae*” revised to “*K. pneumoniae*”.

Line 37, “mainly” revised to “primarily”.

Line 42, “Multi-drug” revised to “Multidrug”.

Line 47, “of” revised to “for”.

Line 50, “during” revised to “from”.

Line 54, “re transmission” revised to “re-transmission”.

Line 58, “multi drug” revised to “multi-drug”.

Line 59, “*Klebsiella pneumoniae*” revised to “*K. pneumoniae*”.

Line 60, “*Escherichia coli*” revised to “*E. coli*”.

Line 64, “*Escherichia coli*” revised to “*E. coli*”.

Line 77, Delete “of the combination”.

Line 78, “residence” revised to “residents,”; “habitat” revised to “habitats”.

Line 82, “isolate” revised to “isolates”.

Line 83, “with phenotype” revised to “with a phenotype”.

Line 84, “carries” revised to “carrying”; “with complex” revised to “with a complex”

Line 85, “*Escherichia coli*” revised to “*E. coli*”.

Line 89, “multi drug” revised to “multi-drug”; “*Escherichia coli*” revised to “*E. coli*”.

Line 90, “multidrug resistant” revised to “multidrug-resistant”.

Line 92, “multidrug resistant” revised to “multidrug-resistant”.

Line 95, “aminoglycosides” revised to “aminoglycosides,”.

Line 96, “*Vibrio spp.*” revised to “*Vibrio spp.*”.

Line 99, “above neglected” revised to “above-neglected”.

Line 104, “*Klebsiella pneumoniae*” revised to “*K. pneumoniae*”.

Line 107, “gene phenotype” revised to “gene-phenotype”.

Line 112, “*Klebsiella pneumoniae*” revised to “*K. pneumoniae*”.

Line 114, “*Klebsiella pneumoniae*” revised to “*K. pneumoniae*”.

Line 115, “*Klebsiella pneumoniae*” revised to “*K. pneumoniae*”.

Line 117, “of MacConkey agar” revised to “of the MacConkey agar”.

Line 123, “issued” revised to “published”.

Line 125. “to” revised to “for the”.

Line 126, “*Escherichia coli*” revised to “*E. coli*”.

Line 131, “Qubit” revised to “Qubit,”.

Line 133, “purification” revised to “purification,”.

Line 135, “canu v1 5” revised to “canu v1.5”. **(This work is completed by biomarker technologies)**

Line 144, “are” revised to “is”.

Line 147, “*Klebsiella pneumoniae*” revised to “*K. pneumoniae*”.

Line 151, “of card” revised to “of the card”; “multi drug” revised to “multi-drug”; “pm911-1.1” revised to “pM911-1.1”.

Line 152, “ps90-2.3” revised to “pS90-2.3”.

Line 155, “gene related” revised to “gene-related”.

Line 161-162, “*Klebsiella pneumoniae*” revised to “*K. pneumoniae*”.

Line 169, “*Klebsiella Pneumoniae*” revised to “*K. pneumoniae*”.

Line 170, “*Klebsiella pneumoniae*” revised to “*K. pneumoniae*”.

Line 171, “antibiotics” revised to “antibiotics,”.

Line 173, “sulfaxazole” revised to “sulfisoxazole”; “*Klebsiella pneumoniae*” revised to “*K. pneumoniae*”.

Line 173-175, “*Klebsiella pneumoniae*” revised to “*K. pneumoniae*”.

Line 177, "*Klebsiella Pneumoniae*" revised to "*K. pneumoniae*".
Line 178, "*Klebsiella Pneumoniae*" revised to "*K. pneumoniae*".
Line 179, "genes" revised to "genes,".
Line 180, Delete "as".
Line 185, "supplementary" revised to "Supplementary".
Line 189, "island" revised to "islands".
Line 190, "supplementary" revised to "Supplementary".
Line 191, "of chromosome" revised to "of the chromosome".
Line 193, Delete "as".
Line 197, "prephage" revised to "prophage"; "supplementary" revised to "Supplementary".
Line 200, "supplementary" revised to "Supplementary".
Line 202, "multidrug" revised to "multi-drug".
Line 204, "*Tral*" revised to "*Tral*,".
Line 207, "any antimicrobial gene was found" revised to "timicrobial gene was found".
Line 211, "containing" revised to "contain".
Line 212, "*Klebsiella pneumoniae*" revised to "*K. pneumoniae*".
Line 213, "*Klebsiella pneumoniae*" revised to "*K. pneumoniae*".
Line 215, "supplementary" revised to "Supplementary".
Line 227, "MDR" revised to "MDR,".
Line 231-232, "*Escherichia fergusonii* plasmid pEF01 in Zhejiang of China, *Klebsiella pneumoniae* plasmid p2018c01-046-3 in Tanwan" revised to "*E. fergusonii* plasmid pEF01 in Zhejiang of China, *K. pneumoniae* plasmid p2018c01-046-3 in Taiwan".
Line 232-233, "*Klebsiella pneumoniae*" revised to "*K. pneumoniae*"; "*Salmonella typhimurium*" revised to "*S. enterica* serovar Typhimurium".
Line 240, "multidrug" revised to "multi-drug".
Line 245, "42.43% ,38 / 94" revised to "42.43%, 38 / 94".
Line 247, "egrets" revised to "egrets,"; "and unknown" revised to "and of unknown".
Line 249, "*Klebsiella pneumoniae*" revised to "*K. pneumoniae*"; "*Escherichia coli*" revised to "*E. coli*".
Line 252, "*Klebsiella pneumoniae*" revised to "*K. pneumoniae*".
Line 253-254, "*Klebsiella pneumoniae*" revised to "*K. pneumoniae*".
Line 256, "*Klebsiella pneumoniae*" revised to "*K. pneumoniae*".
Line 258-262, "*Klebsiella pneumoniae*" revised to "*K. pneumoniae*".
Line 274, "antimicrobial resistant" revised to "antimicrobial-resistant".
Line 277, "*Escherichia coli*" revised to "*E. coli*".
Line 279, "chromosome" revised to "chromosomes"; "*Escherichia coli*" revised to "*E. coli*".
Line 282, Delete "with".
Line 286-287, "can mediate" revised to "it can mediate".
Line 289, "carry resistance" revised to "carry the resistance"; Delete "is".
Line 290, "*Escherichia coli*" revised to "*E. coli*".
Line 293, "*Escherichia coli*" revised to "*E. coli*".

Line 297-298, “Therefore, there may be high level frequent gene exchange among Enterobacteriaceae strains.” revised to “Therefore, there may be a high level of frequent gene exchange among *Enterobacteriaceae* strains.”.

Line 301, “multidrug” revised to “multi-drug”.

Line 303, “indicating” revised to “indicates”.

Line 306, Delete “obviously”.

Line 319, “According to PlasmidFinder database” revised to “According to the PlasmidFinder database”.

Line 320, “group” revised to “group,”.

Line 321, “world,” revised to “world”.

Line 323, “*Escherichia coli*” revised to “*E. coli*”.

Line 325, “repA” revised to “*repA*”.

Line 326, “multidrug” revised to “multi-drug”.

Line 335, “*Escherichia coli*” revised to “*E. coli*”.

Line 336, “downstream are lack of transposase genes” revised to “downstream lack transposase genes”.

Line 344, “transposon mediated” revised to “transposon-mediated”; “transfer” revised to “transfers”.

Line 345, “P1 like” revised to “P1-like”.

Line 346, “*Escherichia coli*” revised to “*E. coli*”.

Line 352, “phage carrying” revised to “phage-carrying”; “ps90-2.3” revised to “pS90-2.3”.

Line 361, “countries,” revised to “countries”.

Line 362, “Laos” revised to “Laos,”.

Line 363, “2020,” revised to “2020”.

Line 364, “ps90-2.3” revised to “pS90-2.3”; “mainly” revised to “primarily”.

Line 367, “Switzerland.” revised to “Switzerland”.

Line 368, “human” revised to “humans”; “may be” revised to “may-be”.

Line 371, “wildbirds” revised to “wild birds”.

Line 382, “multidrug” revised to “multi-drug”.

Line 385, “In a conclusion” revised to “In conclusion”.

Line 387, “multidrug” revised to “multi-drug”

Line 388, “important” revised to “significant”.

Line 389, “AMRS” revised to “AMRs”; “animals” revised to “animals,”.

Line 658, “prephage” revised to “prophage”.

Line 673, “*Klebsiella pneumoniae*” revised to “*K. pneumoniae*”.

Line 694, “*Klebsiella pneumoniae*” revised to “*K. pneumoniae*”.

December 29, 2022

Prof. Chengmin Wang
Guangdong Academy of Sciences
Institute of Zoology, Guangdong Key Laboratory of Animal Conservation and Resource Utilization
Guangzhou, Guangdong
China

Re: Spectrum02691-22R1 (Genomic characteristics and molecular epidemiology of multidrug resistant *Klebsiella pneumoniae* carried by wild birds)

Dear Prof. Chengmin Wang:

Abstract:

The authors should ask a company or a native English speaker for help to improve the language. There are too many grammar problem throughout the whole manuscript. Such as Line 21-22, line 31-34, line 85, and throughout the whole manuscript.

Introduction:

K. pneumoniae was not mentioned and summarized in the introduction, why authors select this species and how about the study of this species in other animals.

Methods:

From which position of the bird the authors isolate *K. pneumoniae*?

Line 140-141: Library construction and Computer sequencing. This is not a sentence, and I do not know what the authors are talking about. Also the next sentence, the whole methods description is cluttered and without any logical.

The software are not cited with reference.

Line 151: "database(<https://cge.cbs.dtu.dk/services/PlasmidFinder/>). the gene information"

Line 164.

In addition, authors did not provide any accession numbers for the genome sequence. The sentence "All accession numbers to bacteria genome or plasmids relating to the paper and deposited in a GenBank database" is with great language problem.

Link Not Available

Sincerely,

Diyan Li

Journals Department
Staff Comments:

Preparing Revision Guidelines

Please return the manuscript within 60 days; if you cannot complete the modification within this time period, please contact me. If you do not wish to modify the manuscript and prefer to submit it to another journal, please notify me of your decision immediately so that the manuscript may be formally withdrawn from consideration by Microbiology Spectrum.

Response to Reviewer Comments

Dear editor and reviewers,

Manuscript ID: Spectrum02691-22 entitled “Genomic characteristics and molecular epidemiology of multidrug resistant *Klebsiella pneumoniae* carried by wild birds” which we submitted to *Microbiology Spectrum* has been reviewed. Thank the reviewers for their valuable comments. Next, we will answer and explain the questions one by one.

Reviewer #1 (Comments for the Author):

The study investigate the genomic features of pathogen bacteria from the wild birds. I have some major concerns:

(1) the title is so large. This study only focus on several species, and 'wild birds' in the title is over-stated.

Responses:

We collected 125 samples from 50 wild bird species including chuckar, red-breasted parakeet, sun parakeet, black-collared starring, japanese white-eye, chestnut-tailed minla, asian barbed owlet, great egret, red-contaminated bulbul, collared scops owl and so on between 2018-2022. 356 pathogenic bacteria of various conditions were isolated from these bird samples which belong to *Enterobacter*, *Aeromonas*, *Proteus*, *Enterococcus* and *Staphylococcus*, among others. Therefore, this study detected chuckar, red-breasted parakeet, sun parakeet, black-collared starring carried *K. pneumoniae* isolates which carried multidrug resistance genes and mobile genetic elements in a large-scale routine screening rather than a few bird species.

(2) what's reason for choose these species?

Responses:

As stated above, we detected the isolates from chuckar were resistant to 9 drugs, the isolates from the red-broken parakeet were resistant to 3 drugs, the isolates from the sun parakeet were resistant to 1 drug, and the isolates from black-colored starring were all sensitive to the drugs used in this study in a routine screen. To assess the association between drug resistance genes and phenotypes, we sequenced the whole genome of the above four *K. pneumoniae* strains and found multiple drug resistance genes and mobile elements on their genomes. It is worth noting that the black-collared starring isolates, although not showing a resistance phenotype, it also carries drug-resistant genes such as β -lactams and tetracyclines.

(3) the samples size is small, and the author only study one bacteria. Thus, the novelty and importance is low.

Responses:

As a common clinical condition pathogen, tens of thousands of human deaths each year are directly related to *K. pneumoniae* infection, which is particularly serious in ICU wards^[1-2]. *K. pneumoniae* is the main carrier of NDM, IPM, and KPC β -lactamase, and widely mediates the spread and transfer of the above resistance genes. At the same time, the bacteria often carry resistance genes of aminoglycosides, tetracyclines, fluoroquinolones and other drugs^[3-5]. Therefore, this strain is the focus of drug resistance research. The mechanism of *Klebsiella* carrying drug resistance genes is relatively complex, which also provides conditions for its resistance to multiple drugs. In our previous research, it was observed that *Klebsiella* carrying multiple replicon plasmids has stronger drug resistance and lower adaptive cost^[6]. In this study, the drug resistance of *K. pneumoniae* carried by wild birds is mainly mediated by plasmids. By querying the NCBI PubMed database, no similar structure and drug resistance gene combination of the plasmid resistant region in this study have been found in wild birds, but this structure has high homology with the drug resistance region carried by human isolated strains. It is difficult to obtain wild animal samples, and most isolates also show drug resistance sensitive phenotype, while the strains involved in this study show strong drug resistance, which should be paid attention to. This study provides new insights into the emergence and prevalence of drug resistance.

References:

- [1] Paterson DL, Ko WC, Von Gottberg A, et al. International prospective study of *Klebsiella pneumoniae* bacteremia: implications of extended-spectrum beta-lactamase production in nosocomial Infections. *Ann Intern Med.* 2004;140(1):26-32.
- [2] Gu D, Dong N, Zheng Z, et al. A fatal outbreak of ST11 carbapenem-resistant hypervirulent *Klebsiella pneumoniae* in a Chinese hospital: a molecular epidemiological study. *Lancet Infect Dis.* 2018;18(1):37-46.
- [3] Wu W, Feng Y, Tang G, et al. NDM Metallo- β -Lactamases and Their Bacterial Producers in Health Care Settings. *Clin Microbiol Rev.* 2019;32(2):e00115-18.
- [4] Wyres KL, Lam MMC, Holt KE. Population genomics of *Klebsiella pneumoniae*. *Nat Rev Microbiol.* 2020;18(6):344-359.
- [5] Grundmann H, Glasner C, Albiger B, et al. Occurrence of carbapenemase-producing *Klebsiella pneumoniae* and *Escherichia coli* in the European survey of carbapenemase-producing *Enterobacteriaceae* (EuSCAPE): a prospective, multinational study. *Lancet Infect Dis.* 2017;17(2):153-163.
- [6] Wang X, Zhao J, Ji F, et al. Multiple-Replicon Resistance Plasmids of *Klebsiella* Mediate Extensive Dissemination of Antimicrobial Genes. *Front Microbiol.*

2021;12:754931.

(4) as I know, now more and more study on the wild birds focus on the virus, which may have the negative effects on the human health. Thus, I don't think this study will bring the interesting for the readers.

Responses:

Wild birds have a high degree of overlap with human activity areas. Wild birds such as sparrows, eurasian blackbird, parrots, egrets and turtledoves are common birds in urban and rural areas, which are significantly affected by human activities. It is well known that the overuse and discharge of antibiotics by humans significantly affect the environmental flora and lead to the evolution of high-level bacterial resistance. The study by Hassell JM, et al. pointed out that non-random processes structure bacterial genetic communities in urban wildlife and demonstrate that communities of avian-borne bacterial genes are shaped by the assemblage of co-existing avian, livestock and human communities, and the habitat within which they exist. The study also proposes a process of drug resistance gene transfer from humans → domestic animals → feces → birds ^[1]. In addition, the migration behavior of birds makes it difficult to control the spread of drug-resistant genes and drug-resistant strains, which poses a serious threat to human health and public health security. For a long time, people may have ignored the important role of wild animals in the epidemic of drug resistance. The important role of wild animals in the epidemic of drug resistance may have been overlooked for a long time. Therefore, it is urgent and important to pay attention to and study the phenomenon of drug resistance transfer caused by wild birds.

References:

[1] Hassell JM, Ward MJ, Muloi D, et al. Deterministic processes structure bacterial genetic communities across an urban landscape. *Nat Commun.* 2019;10(1):2643.

(5) the authors explore the genomic features on this bacteria, and this study may be fit some microbial journal on genomic studies.

Responses:

Our study sequenced the whole genome based on phenotype research to explain the important role of wild birds in the spread of microbial drug resistance by bioinformatics and comparative genomics methods. Genome sequencing and bioinformatics analysis are only a technical means, not the purpose of this study. Identifying the characteristics of genome which is the key link for understanding the origin and evolutionary recombination process of gene fragments mediating drug

resistance. In the field of microbiology, since drug resistance studies are often limited to humans and domestic animals, and wild bird samples are not easily available, the analysis of wild birds accounts for a small proportion of drug-resistant bacteria research. We believe that it is in line with the Microbiology Spectrum magazine's area of interest. Therefore, the key point of this study is to isolate plasmid mediated multidrug resistance *K. pneumoniae* from wild bird samples. Without supervision, the spread of drug resistance mediated by wild birds will cause incalculable losses and risks.

Reviewer #2 (Comments for the Author):

Manuscript ID # Spectrum02691-22 entitled " Genomic characteristics and molecular epidemiology of multidrug resistant *Klebsiella pneumoniae* carried by wild birds" by Xue Wang et al. reports MDR *Klebsiella pneumoniae* were isolated from fresh feces of captured wild bird and showed drug-resistant phenotypes conferred mainly by multidrug-resistant plasmids.

The major comments:

The problem of antibiotic resistance (AMR) caused by antibiotic drug abuse is becoming more and more serious. At present, it has become an important challenge of global public health. The multidrug-resistant *Klebsiella pneumoniae* isolates carried by wild birds had drug-resistant phenotypes conferred mainly by multidrug-resistant plasmids. Wild bird habitats overlap to a greater extent with human and livestock habitats, and further increase the potential for horizontal transfer of multi-drug resistant bacteria between human-animal-environment. This study is very importance to explore the relationship between wild birds and the transmission of multi-drug resistant strain.

Some minor comments:

Line 136, "DNA;" revised to "DNA."

Line 138, ";" revised to "and"

Line 185, "5,374,786kb" should be "5,374,786 b"

Line 211, "5711bp" should be "5,711bp"

Table 1 and Table S1, bird host name should be revised to "English name"

I recommend acceptance after the above minor modifications.

Responses:

Thanks to the reviewer for the suggestion, the above question will be revised.

Line 132, "DNA;" revised to "DNA."

Line 134, ";" revised to "and".

Line 178, "5,374,786kb" revised to "5,374,786bp".

Line 203, "75711bp" revised to "75,711bp".

The host name in Table 1 and Table S1 has been changed to english name. For

details, please refer to the attachments “Table 1” and “Table S1”.

Reviewer #6 (Comments for the Author):

Interesting/relevant; but suggest to enlarge the number and species of animals

Responses:

This study is a long-term survey. From 2018 to 2022, we collected 125 wild bird samples, involving a total of 50 species of birds, such as chuckar, red-breasted parakeet, sun parakeet, black-collared starring, japanese white-eye, etc. 356 strains of various opportunistic pathogens such as *Enterobacter*, *Aeromonas*, *Proteus*, *Enterococcus* and *Staphylococcus* were isolated from these bird samples. Among the 4 isolates involved in this study, 2 were multidrug-resistant bacteria and 2 were sensitive to the drugs we used. In order to compare the genomic differences between multidrug-resistant bacteria and sensitive bacteria, 4 strains of *K. pneumoniae* closely related to human clinic were selected for in-depth research, with a view to answering the important role played by wild birds in the spread of bacterial drug resistance.

Reviewer #7 (Comments for the Author):

This is an interesting manuscript that may be interesting to those who are working in this field. There are multiple typos that need to be fixed. Please see the enclosed marked manuscript.

Responses:

Thanks to the reviewer for the suggestion, the above question will be revised.

Line 22, “*Klebsiella pneumoniae*” revised to “*Klebsiella pneumoniae*”.

Line 24, “was” revised to “were”.

Line 25, “ST629” revised to “ST629,”.

Line 26, “genes” revised to “genes,”; “*bla*_{CTX-M-14} and *bla*_{SHV-11}” revised to “*bla*_{CTX-M-14} and *bla*_{SHV-11}”.

Line 27, “*qacEdelta1*, *sul1* and *aph(3’)-Ib* etc” revised to “*qacEdelta1*, *sul1* and *aph(3’)-Ib* etc”.

Line 28, “*bla*_{SHV-217}” “*bla*_{SHV-1}” “*fosA6*” revised to “*bla*_{SHV-217}” “*bla*_{SHV-1}” “*fosA6*”.

Line 29, “*qnrS1*” “*bla*_{LAP-2}” “*tet(A)*” “*bla*_{SHV-11}” “*fosA6*” revised to “*qnrS1*” “*bla*_{LAP-2}” “*tet(A)*” “*bla*_{SHV-11}” “*fosA6*”.

Line 30, “*tet(A)*” revised to “*tet(A)*”.

Line 35, “*Klebsiella pneumoniae*” revised to “*K. pneumoniae*”.

Line 36, “*Klebsiella pneumoniae*” revised to “*K. pneumoniae*”.

Line 37, “mainly” revised to “primarily”.

Line 42, “Multi-drug” revised to “Multidrug”.

Line 47, “of” revised to “for”.

Line 50, “during” revised to “from”.

Line 54, “re transmission” revised to “re-transmission”.

Line 58, “multi drug” revised to “multi-drug”.

Line 59, “*Klebsiella pneumoniae*” revised to “*K. pneumoniae*”.

Line 60, “*Escherichia coli*” revised to “*E. coli*”.

Line 64, “*Escherichia coli*” revised to “*E. coli*”.

Line 77, Delete “of the combination”.

Line 78, “residence” revised to “residents,”; “habitat” revised to “habitats”.

Line 82, “isolate” revised to “isolates”.

Line 83, “with phenotype” revised to “with a phenotype”.

Line 84, “carries” revised to “carrying”; “with complex” revised to “with a complex”

Line 85, “*Escherichia coli*” revised to “*E. coli*”.

Line 89, “multi drug” revised to “multi-drug”; “*Escherichia coli*” revised to “*E. coli*”.

Line 90, “multidrug resistant” revised to “multidrug-resistant”.

Line 92, “multidrug resistant” revised to “multidrug-resistant”.

Line 95, “aminoglycosides” revised to “aminoglycosides,”.

Line 96, “*Vibrio spp.*” revised to “*Vibrio spp.*”.

Line 99, “above neglected” revised to “above-neglected”.

Line 104, “*Klebsiella pneumoniae*” revised to “*K. pneumoniae*”.

Line 107, “gene phenotype” revised to “gene-phenotype”.

Line 112, “*Klebsiella pneumoniae*” revised to “*K. pneumoniae*”.

Line 114, “*Klebsiella pneumoniae*” revised to “*K. pneumoniae*”.

Line 115, “*Klebsiella pneumoniae*” revised to “*K. pneumoniae*”.

Line 117, “of MacConkey agar” revised to “of the MacConkey agar”.

Line 123, “issued” revised to “published”.

Line 125. “to” revised to “for the”.

Line 126, “*Escherichia coli*” revised to “*E. coli*”.

Line 131, “Qubit” revised to “Qubit,”.

Line 133, “purification” revised to “purification,”.

Line 135, “canu v1 5” revised to “canu v1.5”. **(This work is completed by biomarker technologies)**

Line 144, “are” revised to “is”.

Line 147, “*Klebsiella pneumoniae*” revised to “*K. pneumoniae*”.

Line 151, “of card” revised to “of the card”; “multi drug” revised to “multi-drug”; “pm911-1.1” revised to “pM911-1.1”.

Line 152, “ps90-2.3” revised to “pS90-2.3”.

Line 155, “gene related” revised to “gene-related”.

Line 161-162, “*Klebsiella pneumoniae*” revised to “*K. pneumoniae*”.

Line 169, “*Klebsiella Pneumoniae*” revised to “*K. pneumoniae*”.

Line 170, “*Klebsiella pneumoniae*” revised to “*K. pneumoniae*”.

Line 171, “antibiotics” revised to “antibiotics,”.

Line 173, “sulfaxazole” revised to “sulfisoxazole”; “*Klebsiella pneumoniae*” revised to “*K. pneumoniae*”.

Line 173-175, “*Klebsiella pneumoniae*” revised to “*K. pneumoniae*”.

Line 177, "*Klebsiella Pneumoniae*" revised to "*K. pneumoniae*".

Line 178, "*Klebsiella Pneumoniae*" revised to "*K. pneumoniae*".

Line 179, "genes" revised to "genes,".

Line 180, Delete "as".

Line 185, "supplementary" revised to "Supplementary".

Line 189, "island" revised to "islands".

Line 190, "supplementary" revised to "Supplementary".

Line 191, "of chromosome" revised to "of the chromosome".

Line 193, Delete "as".

Line 197, "prephage" revised to "prophage"; "supplementary" revised to "Supplementary".

Line 200, "supplementary" revised to "Supplementary".

Line 202, "multidrug" revised to "multi-drug".

Line 204, "*Tral*" revised to "*Tral*,".

Line 207, "any antimicrobial gene was found" revised to "timicrobial gene was found".

Line 211, "containing" revised to "contain".

Line 212, "*Klebsiella pneumoniae*" revised to "*K. pneumoniae*".

Line 213, "*Klebsiella pneumoniae*" revised to "*K. pneumoniae*".

Line 215, "supplementary" revised to "Supplementary".

Line 227, "MDR" revised to "MDR,".

Line 231-232, "*Escherichia fergusonii* plasmid pEF01 in Zhejiang of China, *Klebsiella pneumoniae* plasmid p2018c01-046-3 in Tanwan" revised to "*E. fergusonii* plasmid pEF01 in Zhejiang of China, *K. pneumoniae* plasmid p2018c01-046-3 in Taiwan".

Line 232-233, "*Klebsiella pneumoniae*" revised to "*K. pneumoniae*"; "*Salmonella typhimurium*" revised to "*S. enterica* serovar Typhimurium".

Line 240, "multidrug" revised to "multi-drug".

Line 245, "42.43% ,38 / 94" revised to "42.43%, 38 / 94".

Line 247, "egrets" revised to "egrets,"; "and unknown" revised to "and of unknown".

Line 249, "*Klebsiella pneumoniae*" revised to "*K. pneumoniae*"; "*Escherichia coli*" revised to "*E. coli*".

Line 252, "*Klebsiella pneumoniae*" revised to "*K. pneumoniae*".

Line 253-254, "*Klebsiella pneumoniae*" revised to "*K. pneumoniae*".

Line 256, "*Klebsiella pneumoniae*" revised to "*K. pneumoniae*".

Line 258-262, "*Klebsiella pneumoniae*" revised to "*K. pneumoniae*".

Line 274, "antimicrobial resistant" revised to "antimicrobial-resistant".

Line 277, "*Escherichia coli*" revised to "*E. coli*".

Line 279, "chromosome" revised to "chromosomes"; "*Escherichia coli*" revised to "*E. coli*".

Line 282, Delete "with".

Line 286-287, "can mediate" revised to "it can mediate".

Line 289, "carry resistance" revised to "carry the resistance"; Delete "is".

Line 290, "*Escherichia coli*" revised to "*E. coli*".

Line 293, "*Escherichia coli*" revised to "*E. coli*".

Line 297-298, “Therefore, there may be high level frequent gene exchange among Enterobacteriaceae strains.” revised to “Therefore, there may be a high level of frequent gene exchange among *Enterobacteriaceae* strains.”.

Line 301, “multidrug” revised to “multi-drug”.

Line 303, “indicating” revised to “indicates”.

Line 306, Delete “obviously”.

Line 319, “According to PlasmidFinder database” revised to “According to the PlasmidFinder database”.

Line 320, “group” revised to “group,”.

Line 321, “world,” revised to “world”.

Line 323, “*Escherichia coli*” revised to “*E. coli*”.

Line 325, “repA” revised to “*repA*”.

Line 326, “multidrug” revised to “multi-drug”.

Line 335, “*Escherichia coli*” revised to “*E. coli*”.

Line 336, “downstream are lack of transposase genes” revised to “downstream lack transposase genes”.

Line 344, “transposon mediated” revised to “transposon-mediated”; “transfer” revised to “transfers”.

Line 345, “P1 like” revised to “P1-like”.

Line 346, “*Escherichia coli*” revised to “*E. coli*”.

Line 352, “phage carrying” revised to “phage-carrying”; “ps90-2.3” revised to “pS90-2.3”.

Line 361, “countries,” revised to “countries”.

Line 362, “Laos” revised to “Laos,”.

Line 363, “2020,” revised to “2020”.

Line 364, “ps90-2.3” revised to “pS90-2.3”; “mainly” revised to “primarily”.

Line 367, “Switzerland.” revised to “Switzerland”.

Line 368, “human” revised to “humans”; “may be” revised to “may-be”.

Line 371, “wildbirds” revised to “wild birds”.

Line 382, “multidrug” revised to “multi-drug”.

Line 385, “In a conclusion” revised to “In conclusion”.

Line 387, “multidrug” revised to “multi-drug”

Line 388, “important” revised to “significant”.

Line 389, “AMRS” revised to “AMRs”; “animals” revised to “animals,”.

Line 658, “prephage” revised to “prophage”.

Line 673, “*Klebsiella pneumoniae*” revised to “*K. pneumoniae*”.

Line 694, “*Klebsiella pneumoniae*” revised to “*K. pneumoniae*”.

Response to Reviewer Comments

Dear editor,

Manuscript ID: Spectrum02691-22R2 entitled “Genomic characteristics and molecular epidemiology of multidrug resistant *Klebsiella pneumoniae* carried by wild birds” which we submitted to *Microbiology Spectrum* has been reviewed. Thank the reviewers for their valuable comments. Next, we will answer and explain the questions one by one.

1. The authors should ask a company or a native English speaker for help to improve the language. There are too many grammar problems throughout the whole manuscript. Such as Line 21-22, line 31-34, line 85, and throughout the whole manuscript.

Responses:

We will entrust American Journal Experts (AJE) to polish and modify the article to improve the language level.

2. Introduction:

K. pneumoniae was not mentioned and summarized in the introduction, why authors select this species and how about the study of this species in other animals.

Responses:

In the introduction, we added the background material of selecting *Klebsiella pneumoniae* as the research object.

Line 53-60, Add “*Klebsiella pneumoniae* has long been considered a pathogen and is still one of the most common nosocomial pathogens in the world (1). It is widespread in Asia, Africa, and Europe, causing tens of thousands of infections and deaths every year (2, 3, 4). Notably, *K. pneumoniae* can occupy favorable niches in plants, animals, and the environment. Some studies have noted that *K. pneumoniae* can infect California sea lions and African green monkeys, causing invasive pneumonia (5, 6). It is also a common pathogen causing cow mastitis (7), and human clinical isolates share similar characteristics with strains from other sources (8). The World Health Organization recognizes extended-spectrum β -lactam producing and carbapenem-resistant *K. pneumoniae* as a critical public health threat (9).”

Line 422-450, Added references 1-9, and push back the rest of the reference numbers in order.

1. Pendleton JN, Gorman SP, Gilmore BF. 2013. Clinical relevance of the ESKAPE pathogens. *Expert Rev Anti Infect Ther.* 11(3):297-308.

2. Cassini A, Högberg LD, Plachouras D, Quattrocchi A, Hoxha A, Simonsen GS, Colomb-Cotinat M, Kretzschmar ME, Devleeschauwer B, Cecchini M, Ouakrim DA,

Oliveira TC, Struelens MJ, Suetens C, Monnet DL. 2019. Burden of AMR Collaborative Group. Attributable deaths and disability-adjusted life-years caused by infections with antibiotic-resistant bacteria in the EU and the European Economic Area in 2015: a population-level modelling analysis. *The Lancet. Infectious diseases*. 19(1):56-66.

3. Musicha P, Cornick JE, Bar-Zeev N, French N, Masesa C, Denis B, Kennedy N, Mallewa J, Gordon MA, Msefula CL, Heyderman RS, Everett DB, Feasey NA. 2017. Trends in antimicrobial resistance in bloodstream infection isolates at a large urban hospital in Malawi (1998-2016): a surveillance study. *The Lancet. Infectious diseases*. 17(10):1042-1052.

4. Gandra S, Alvarez-Uria G, Turner P, Joshi J, Limmathurotsakul D, van Doorn HR. 2020. Antimicrobial Resistance Surveillance in Low- and Middle-Income Countries: Progress and Challenges in Eight South Asian and Southeast Asian Countries. *Clinical microbiology reviews*. 33(3): e00048-19.

5. Jang S, Wheeler L, Carey RB, Jensen B, Crandall CM, Schrader KN, Jessup D, Colegrove K, Gulland FM. 2010. Pleuritis and suppurative pneumonia associated with a hypermucoviscosity phenotype of *Klebsiella pneumoniae* in California sea lions (*Zalophus californianus*). *Veterinary microbiology*. 141(1-2):174-7.

6. Twenhafel NA, Whitehouse CA, Stevens EL, Hottel HE, Foster CD, Gamble S, Abbott S, Janda JM, Kreiselmeier N, Steele KE. 2008. Multisystemic abscesses in African green monkeys (*Chlorocebus aethiops*) with invasive *Klebsiella pneumoniae*--identification of the hypermucoviscosity phenotype. *Veterinary pathology*. 45(2):226-31.

7. Schukken Y, Chuff M, Moroni P, Gurjar A, Santisteban C, Welcome F, Zadoks R. 2012. The "other" gram-negative bacteria in mastitis: *Klebsiella*, *serratia*, and more. *The Veterinary clinics of North America. Food animal practice*. 28(2):239-56.

8. Struve C, Krogfelt KA. 2004. Pathogenic potential of environmental *Klebsiella pneumoniae* isolates. *Environmental microbiology*. 6(6):584-90.

9. Breijyeh Z, Jubeh B, Karaman R. 2020. Resistance of Gram-Negative Bacteria to Current Antibacterial Agents and Approaches to Resolve It. *Molecules (Basel,*

Switzerland). 25(6):1340.

3.Methods:

From which position of the bird the authors isolate *K. pneumoniae*?

Responses:

We isolated *Klebsiella pneumoniae* from the feces of wild birds.

Line 126-128, “*K. pneumoniae* strains were isolated from feces of smuggled wild birds, including Chukar Partridge (*Alectoris chukar*), Red-breasted Parakeet (*Psittacula alexandri*), Sun Parakeet (*Aratinga solstitialis*) and Black-collared Starling (*Sturnus nigricollis*).”

4.Line 140-141: Library construction and Computer sequencing. This is not a sentence, and I do not know what the authors are talking about. Also the next sentence, the whole methods description is cluttered and without any logical.

The software are not cited with reference.

Responses:

We used Oxford nanopore sequencing technology to sequence the whole genome. After obtaining the data, we used the software canu v1.5 for assembly and the software Pilon for correction. All the above work is completed by Biomarker Technologies (China).

Line 148-152, “After obtaining the data, the subreads with low quality and those that were too short were filtered, and canu v1.5 software was used to reassemble the filtered subreads from scratch. The draft genome was assembled with Pilon software. The genomic DNA library was constructed, and whole genome sequencing was performed, with an estimated size of 6 Mbp. The sequencing depth was $\geq 100x$, 0 gap.”

5.Line 151: "database(<https://cge.cbs.dtu.dk/services/PlasmidFinder/>). the gene information"

Responses:

We use the PlasmidFinder database to determine the type of plasmid incompatibility group. The service now points to the link <https://bitbucket.org/genomicepidemiology/plasmidfinder>.

Line 156-158, “The incompatible group types of plasmids were analyzed by using plasma finder database(<https://cge.cbs.dtu.dk/services/PlasmidFinder/>).” revised to “The incompatible group types of plasmids were analyzed by using the PlasmidFinder database (<https://bitbucket.org/genomicepidemiology/plasmidfinder>).”

We use CARD database to predict drug resistance genes.

Line 158-159, “the gene information related to antibiotic resistance is determined by using the bioinformatics database card (the comprehensive antimicrobial resistance

database (<https://card.mcmaster.ca/home>)." revised to "The comparative antibiotic resistance database (CARD, <https://card.mcmaster.ca/home>) was used to annotate the genes related to drug resistance."

6.Line 164.

Responses:

Line 175, "*Construction of Bacterial Genome-Wide Phylogenetic Tree*" revised to "*Construction of the whole genome phylogenetic tree of K. pneumoniae*".

7.In addition, authors did not provide any accession numbers for the genome sequence. The sentence "All accession numbers to bacteria genome or plasmids relating to the paper and deposited in a GenBank database" is with great language problem.

Responses:

All login numbers submitted to the Genbank database are displayed in the supplementary materials. See Table S1 and Table S2 for details.

Line 417-419, "All accession numbers to bacteria genome or plasmids relating to the paper and deposited in a GenBank database." revised to "All accession numbers for bacterial genomes or plasmids related to the paper were deposited in a GenBank database (Table S1 and Table S2). All the data are available in the main text or supplementary materials."

In addition, the following changes have been made to the content of the text:

Line1-2, "Genomic characteristics and molecular epidemiology of multidrug resistant *Klebsiella pneumoniae* carried by wild birds" revised to "Genomic characteristics and molecular epidemiology of multidrug-resistant *Klebsiella pneumoniae* strains carried by wild birds".

Line 16, "*Equally to this work." revised to "*Contributed equally to this work."

Line 21, "multi-drug" revised to "multidrug".

Line 22, "were" revised to "was".

Line 22, Delete "were".

Line 23, "micro-broth" revised to "the microbroth".

Line 24, "all" revised to "both".

Line 24, Add "and".

Line 25, "but" revised to "while".

Line 25, "was" revised to "were".

Line 27, Delete "etc".

Line 28, Delete "type".

Line 30, Delete "The".

Line 32, “Enterobacteriaceae strain” revised to “*Enterobacteriaceae* strain”.

Line 35, “human bacteria hosts in China” revised to “isolated human-infecting bacteria in China”.

Line 35, “whose are” revised to “namely,”.

Line 37, “drug-resistant” revised to “drug resistance”.

Line 37, “multidrug-resistant” revised to “multidrug resistance”.

Line 37, “which” revised to “that”.

Line 38, “human bacteria hosts” revised to “human-infecting bacteria”.

Line 41, “Until now,” revised to “Little”.

Line 44, “human bacterial hosts.” revised to “human-infecting bacteria.”.

Line 44, “great” revised to “great”.

Line 45, “and” revised to “which”.

Line 45, “increase” revised to “increases”.

Line 45, “multi-drug” revised to “multidrug”.

Line 46, “between human-animal-environment” revised to “among humans, animals, and the environment”.

Line 47, “multi drug resistant” revised to “multidrug-resistant”.

Line 47, “paid attention to and supervised” revised to “given attention and monitored”.

Line 49, “Drug-resistant plasmids” revised to “Drug resistance plasmids”

Line 61, “more and more” revised to “increasingly”.

Line 62, “has become” revised to “represents”.

Line 62, “is” revised to “has been”.

Line 64, “dose” revised to “doses”.

Line 65, “even reached more than” revised to “exceeded”.

Line 66, “number” revised to “amount”.

Line 67, “are” revised to “is”.

Line 67, “to” revised to “in”.

Line 67-68, “by fertilizing land runoff” revised to “via runoff from fertilized land”.

Line 68, “which will lead” revised to “leading”.

Line 68, “re-transmission” revised to “retransmission”.

Line 68, “drug-resistant genes” revised to “drug resistance genes”.

Line 69, “gene” revised to “genetic”.

Line 71, “making” revised to “attenuating”.

Line 71, Delete “increasingly attenuated”.

Line 72, “in zoo” revised to “in a zoos”.

Line 72, “circle carry multi-drug resistant” revised to “carried the multidrug-resistant”.

Line 73, “carry” revised to “carried”.

Line 74, Add “which are”.

Line 74, Add “the”.

Line 75, “A” revised to “a”.

Line 76, “carbapenemase” revised to “the carbapenemase genes”.

Line 77, Add “the”.

Line 77, “which” revised to “and”.

Line 77, Add “the”.

Line 78, “of” revised to “from a”.

Line 79, “carries” revised to “carried”.

Line 79, “carries” revised to “carried”.

Line 82, “More” revised to “Several”.

Line 82, “host” revised to “host-derived”.

Line 82, “are carried” revised to “carrying”.

Line 83, Add “resistance to”.

Line 85, “overlapping areas” revised to “areas of overlap”.

Line 85, “the” revised to “their”.

Line 86, “Its” revised to “Their”.

Line 86, “distance” revised to “distances”.

Line 87, “it” revised to “wild birds”.

Line 87, “AMR strains” revised to “strains with AMR”.

Line 87, “About” revised to “Approximately”.

Line 88, “The” revised to “A”.

Line 89, “ARGs” revised to “ARG”.

Line 89, “when there are” revised to “in the presence of”.

Line 90, “increases” revised to “increased”.

Line 90, “the increase of” revised to “increasing”.

Line 91, Add “strains derived from”.

Line 92, “bird habitats” revised to “birds in the habitat”.

Line 92, Add “genetics of the”.

Line 92, “genetic community” revised to “community”.

Line 93, “forms” revised to “constitute”.

Line 93, “non-random” revised to “nonrandom”.

Line 93, “,” revised to “.”.

Line 95, Delete “Australian wild gull (*Chrococephalus novaehollandiae*)”.

Line 95, “a” revised to “A”.

Line 96, “SG17-135” revised to “(SG17-135)”.

Line 96, Add “resistance to”.

Line 97, Add “the”.

Line 99, Add “was identified from Australian wild gull (*Chrococephalus novaehollandiae*)”.

Line 100, Add “strains”.

Line 100, “the wild” revised to “wild”.

Line 100, “the white” revised to “white”.

Line 101, Delete “the”.

Line 101, “gene” revised to “genes”.

Line 101, “while the cattle” revised to “and the cattle”.

Line 102, “carried” revised to “carry”.

Line 103, “the” revised to “a”.

Line 103, “ β - Among the lactam” revised to “ β -lactam”.

Line 106, Add “the”.

Line 107, “The *Vibrio* spp.” revised to “*Vibrio* spp.”.

Line 108, “multiple drug” revised to “a multidrug”.

Line 108-109, “resistant drugs” revised to “drugs associated with the phenotypes”.

Line 109, “it” revised to “the study”.

Line 111, Add “strains carried by”.

Line 111, Add “strains”.

Line 112, “is” revised to “were”.

Line 112, “and has” revised to “, had an”.

Line 112, “carries” revised to “carried the”.

Line 113, “The above-neglected” revised to “Neglect of the above”.

Line 114, “with” revised to “via”.

Line 114, “gene” revised to “genetic”.

Line 115, “drug-resistant genes” revised to “drug resistance genes”.

Line 116, “to reveal” revised to “for revealing”.

Line 117, “AMRS” revised to “AMR strains”.

Line 117, “establish” revised to “establishing”.

Line 120, “technology and” revised to “and”.

Line 121, “was” revised to “were”.

Line 125, “*Strains isolation*” revised to “*Strain isolation*”.

Line 126, “smuggling feces of” revised to “feces of smuggled”.

Line 128, Add “isolated”.

Line 128, Add “a”.

Line 130, Add “isolated”.

Line 130, Add “a”.

Line 131, “province” revised to “Province”.

Line 136, “test” revised to “testing guidelines”.

Line 137, “test” revised to “testing”.

Line 138-140, “Cefuroxime, Ceftriaxone, Cefepime, Ampicillin/Sulbactam, Piperacillin/Tazobactam, Meropenem, Gentamicin, Amikacin and Chloramphenicol were used for the Antibiotic susceptibility test by the Minimum Inhibitory Concentration” revised to “cefuroxime, ceftriaxone, cefepime, ampicillin/sulbactam, piperacillin/tazobactam, meropenem, gentamicin, amikacin and chloramphenicol were used for antibiotic susceptibility testing at the minimum inhibitory concentration”.

Line 143, Add “was performed”.

Line 143, “nanopore” revised to “the Nanopore”.

Line 143-144, “biomarker technologies, China” revised to “Biomarker Technologies, China”.

Line 145, Add “a”.

Line 145, Add “a”.

Line 145, Add “The”.

Line 146, Add “was performed”.

Line 146-148, “(sqk-lsk109 ligation kit, including DNA damage repair and terminal repair, junction connection, magnetic bead purification, and qubit library quantification)” revised to “sqk-lsk109 ligation kit (including DNA damage repair and terminal repair, junction connection, magnetic bead purification, and qubit library quantification)”.

Line 148 “and the library was subjected to sequencing.” revised to “and the library was subjected to sequencing.”.

Line 149, “filter the” revised to “the”.

Line 149, Add “those that were”.

Line 149, “length” revised to “were filtered”.

Line 149-152, “use the software canu v1.5 reassemble the filtered subreads from scratch, correct the assembled draft genome with the software Pilon, construct the genomic DNA library, and complete the sequencing of the bacteria with an estimated size of 6m. The sequencing depth is $\geq 100x$, 0 gap.” revised to “canu v1.5 software was used to reassemble the filtered subreads from scratch. The draft genome was assembled with Pilon software. The genomic DNA library was constructed, and whole genome sequencing was performed, with an estimated size of 6 Mbp. The sequencing depth was $\geq 100x$, 0 gap.”.

Line 154, “are” revised to “were”.

Line 154, Add “the”.

Line 154, “,” revised to “databases, the”.

Line 155, “TN number registry(<https://transposon.lstmed.ac.uk/>)” revised to “(<https://transposon.lstmed.ac.uk/>)”.

Line 155, Add “the”.

Line 156, Add “the”.

Line 157, Add “the”.

Line 158, “Use the” revised to “The”.

Line 159, Add “was used”.

Line 160, “Further” revised to “Furthermore”.

Line 160, “are” revised to “were”.

Line 161, “typing by” revised to “by”.

Line 161, Add “the”.

Line 161, “typing database” revised to “database”.

Line 162, Add “the”.

Line 162, “are” revised to “were”.

Line 166, “card” revised to “CARD”.

Line 166, “multi-drug resistant” revised to “the multidrug resistance”.

Line 168, “NCBI was used to blast” revised to “The NCBI database was used for blast analysis of”.

Line 169, “of” revised to “in”.

Line 170, “Through” revised to “The”.

Line 171, “Search” revised to “was used to search”.

Line 172, “are” revised to “were”.

Line 172, “collect their” revised to “their”.

Line 173, Add “were collected”.

Line 174, “realized” revised to “performed”.

Line 174, “software mauve” revised to “mauve software”.

Line 175, “*whole*” revised to “*the whole*”.

Line 176, “similar” revised to “for similar”.

Line 176, “PLSDB database” revised to “PLSDB”.

Line 177, “MLST typing” revised to “MLST”.

Line 184, “phenotype” revised to “phenotypes”.

Line 184, “was” revised to “were”.

Line 185, “multiple resistance” revised to “resistance”.

Line 185, “including” revised to “namely”.

Line 188, “The *K. pneumoniae*” revised to “*K. pneumoniae*”.

Line 189, “All the 4” revised to “All 4”.

Line 192, Add “the”.

Line 192, “is” revised to “was”.

Line 192, “5,374,786bp” revised to “5,374,786 bp”.

Line 193, “genome carries” revised to “carried”.

Line 194, “110,388bp” revised to “110,388 bp”.

Line 195, “109,675bp” revised to “109,675 bp”.

Line 195, “/ IncFII” revised to “/IncFII”.

Line 196, “*mphA*, *dfrA12*, *aadA2*, *qacEdelta1*, *sul1*, *tet(A)*, *aph(3')-Ia*, *sul2*”
revised to “*mphA*, *dfrA12*, *aadA2*, *qacEdelta1*, *sul1*, *tet(A)*, *aph(3')-Ia*, *sul2*”.

Line 197, “*aph(3')-Ib*,” revised to “*aph(3')-Ib*”.

Line 198, “contain” revised to “contained”.

Line 198, “contain” revised to “contained”.

Line 198, “prophage, respectively” revised to “prophage each”.

Line 200, “is” revised to “was”.

Line 200, “carries” revised to “carried”.

Line 202, “plasmids” revised to “plasmids”.

Line 202, “as pS141.1” revised to “pS141.1”.

Line 202, “112,160bp” revised to “112,160 bp”.

Line 203, “contains” revised to “contained”.

Line 203, “islands” revised to “islands,”.

Line 203, “has a” revised to “had one”.

Line 204, “carries” revised to “carried”.

Line 204, “-resistant” revised to “resistance-related”.

Line 204, “gene” revised to “gene,”.

Line 205, “is” revised to “was”.

Line 205, “192bp” revised to “192 bp”.

Line 206, “type that” revised to “that was”.

Line 206, Add “the”.

Line 206, “carries” revised to “carried”.

Line 208, “75,711bp” revised to “75,711 bp”.

Line 208, “21,377bp” revised to “21,377 bp”.

Line 210, “contains” revised to “contained”.

Line 210, “has” revised to “had”.

Line 211, “has” revised to “had”.

Line 212, “is” revised to “was”.

Line 212, “carries” revised to “carried”.

Line 213, “carries” revised to “carried”.

Line 214, “150,355bp” revised to “150,355 bp”.

Line 214, “carries” revised to “carried”.

Line 215, “Islands” revised to “islands”.

Line 217, “*multi-drug*” revised to “*the multidrug*”.

Line 218, “has” revised to “had”.

Line 218, “75,711bp” revised to “75,711 bp”.

Line 218, “There are” revised to “It contained”.

Line 219, “*trbB, traY, traL, traI, and traJ*” revised to “*trbB, traY, traL, traI, and traJ*”.

Line 219, “3- 35,601bp” revised to “3-35,601 bp”.

Line 219, Add “the”.

Line 220, “are” revised to “were”.

Line 221, “40,687 - 45,449bp” revised to “40,687-45,449 bp”.

Line 222, “timicrobial” revised to “antimicrobial”.

Line 222, Add “resistance”.

Line 222, “upstream and” revised to “upstream or”.

Line 223, “PlasmaFinder” revised to “PlasmidFinder”.

Line 223, “incompatible” revised to “incompatibility”.

Line 223, Add “that of”.

Line 224, “patible” revised to “patibility”.

Line 226, “104bp” revised to “104 bp”.

Line 226, “50,659 – 63,762bp” revised to “50,659–63,762 bp”.

Line 227, “contain” revised to “containing”.

Line 227, Add “, namely”.

Line 227, Add “,”.

Line 228, “was are” revised to “was”.

Line 228, “with” revised to “to the”.

Line 229, “Zhejiang of” revised to “Zhejiang”.

Line 229-230, “*Raoultella ornithinolytica* plasmids” revised to “the *Raoultella ornithinolytica* plasmid”.

Line 230, “Tokyo of Japan” revised to “Tokyo, Japan”.

Line 230, Add “the”.

Line 231, “Swiss” revised to “Switzerland”.

Line 231, “a length 1,180bp” revised to “a 1,180 bp”.

Line 231-232, “643bp” revised to “643 bp”.

Line 232, Add “the”.

Line 232, “contains” revised to “contained”.

Line 233, “is” revised to “carried the”.

Line 233, Add “the”.

Line 235, “60,925–62,124bp” revised to “60,925–62,124 bp”.

Line 237, Add “with”.

Line 237, “57,825bp” revised to “57,825 bp”.

Line 237, “belongs” revised to “belonged”.

Line 237, Add “the”.

Line 237, “incompatible” revised to “incompatibility”.

Line 238, “47,134-52,200bp” revised to “47,134-52,200 bp”.

Line 238, “contains” revised to “contained”.

Line 239, “is” revised to “was”.

Line 240, “19,709- 56,370bp” revised to “19,709-56,370 bp”.

Line 240-241, “56,431-57,234bp” revised to “56,431-57,234 bp”.

Line 242, “are” revised to “were”.

Line 243, “is” revised to “was”.

Line 243, Add “regions”.

Line 243, Add “the”.

Line 244, “13,626bp” revised to “13,626 bp”.

Line 244, “40,003bp” revised to “40,003 bp”.

Line 244, Add “the”.

Line 245, “15,907bp” revised to “15,907 bp”.

Line 245, “57,234bp” revised to “57,234 bp”.

Line 246, “32,359- 40,003” revised to “32,359-40,003”.

Line 246, Add “the”.

Line 247, “is” revised to “was”.

Line 247, Add “the”.

Line 247, Add “,”.

Line 248, “of China” revised to “China”.

Line 248, “of China” revised to “China”.

Line 250, “is” revised to “was”.

Line 250, “genomes” revised to “sequences”.

Line 251, Add “the”.

Line 252, Add “the”.

Line 252, Add “the”.

Line 252-253, “39,299- 40,003bp” revised to “39,299-40,003 bp”.

Line 253, “but” revised to “while”.

Line 253, Add “the”.

Line 254, “32,359 -32,709bp” revised to “32,359-32,709 bp”.

Line 254, “39,299-40,003bp” revised to “39,299-40,003 bp”.

Line 257, “*multi-drug resistant*” revised to “*multidrug resistance*”.

Line 258, “there are 61” revised to “61 of the”.

Line 259, Add “screened were”.

Line 259, “32” revised to “Thirty-two”.

Line 260, Add “were”.

Line 260, "(42 / 61)" revised to "(42/61)".

Line 260-261, "human host bacteria" revised to "human-infecting bacteria".

Line 261, "other" revised to "others".

Line 261-262, "host bacteria of pigs and unknown origin (Figure 4A)" revised to "bacteria derived from pig hosts and those of unknown origin (Figure 4A)".

Line 262, "94" revised to "Ninety-four".

Line 262-263, "they were also mainly isolated in 2020 (42.43%, 38 / 94)" revised to "most isolates were obtained in 2020 (42.43%, 38/94)".

Line 263, "(16 / 94)" revised to "(16/94) were".

Line 263, Add "A total of".

Line 263, "(34 / 94)" revised to "(34/94) were".

Line 264, "human host bacteria" revised to "human-infecting bacteria".

Line 264, "other" revised to "others were".

Line 264, "from host bacteria of dogs," revised to "from bacteria infecting dogs,".

Line 264-265, ", egrets" revised to ", and egrets".

Line 265, Add "those".

Line 265, "are" revised to "were".

Line 266, "gene structure" revised to "structural gene".

Line 266-267, "composition is" revised to "compositions were".

Line 267, "remains" revised to "remained".

Line 267, "take" revised to "had".

Line 268, "bacteria" revised to "bacterial".

Line 271-272, "carry similar plasmid (similarity \geq 95%) with pM911-1.1" revised to "carrying similar plasmids (similarity \geq 95%) to pM911-1.1".

Line 272, Add "the".

Line 273, Add "a".

Line 273, Add "a".

Line 273, Add ",".

Line 273, Add ", were used".

Line 274. Add "a".

Line 274, "base the" revised to "based on the".

Line 274, "bacteria genome" revised to "bacterial genome sequences".

Line 275, "ST11 type and located in" revised to "ST11 and were located in".

Line 276, "the six" revised to "six".

Line 276, “the four” revised to “four”.

Line 277-278, “and *K. pneumoniae*” revised to “*K. pneumoniae*”.

Line 278, Add “of”.

Line 278, Add “origin,”.

Line 278, “are” revised to “were”.

Line 279, “the *K. pneumoniae*” revised to “*K. pneumoniae*”.

Line 279, Add “a”.

Line 279, “is” revised to “was”.

Line 280, “the *Klebsiella pneumoniae*” revised to “*Klebsiella pneumoniae*”.

Line 281, “is located” revised to “was located”.

Line 281, “the SCKP020135” revised to “SCKP020135”.

Line 282, “is in” revised to “was in”.

Line 282, Add “a”.

Line 288, “ESBL” revised to “ESBLs”.

Line 288, “mediated by” revised to “encoded by”.

Line 290, Add “strains”.

Line 292, “genome” revised to “genomes”.

Line 293, Delete “gene”.

Line 293, “about” revised to “approximately”.

Line 293, Add “the”.

Line 294, “structure was similar” revised to “structures are similar”.

Line 295, Add “the”.

Line 296, “was” revised to “is”.

Line 296, “and less” revised to “strain and fewer”.

Line 297, Add “strains”.

Line 297, “the” revised to “a”.

Line 298, “drugs” revised to “drug”.

Line 302, “late” revised to “at a later stage”.

Line 302, Add “available for this gene.”.

Line 304, “Resistance” revised to “ampicillin resistance”.

Line 304, “of ampicillin” revised to “observed”.

Line 306, “carry” revised to “carried”.

Line 307, Add “the”.

Line 308, “the *FosA6*” revised to “*fosA6*”.

Line 310, “mediated” revised to “exhibited”.

Line 310, “fosomycin” revised to “fosfomycin”.

Line 311, “shows” revised to “showed”.

Line 311, Add “the”.

Line 311, “come” revised to “have been derived”.

Line 311, “and mediated” revised to “mediated”.

Line 313, Add “the”.

Line 313-314, “ β - lactams and aminoglycosides” revised to “ β -lactam and aminoglycoside”.

Line 314, “there may be a high level of frequent” revised to “frequent”.

Line 314, Add “may occur”.

Line 317, “prophages, etc.” revised to “prophages”.

Line 317-318, “multi-drug” revised to “the multidrug”.

Line 318, “plasmid” revised to “plasmids”.

Line 318, “are” revised to “were”.

Line 320, “indicates” revised to “indicating”.

Line 321, “is” revised to “was”.

Line 321, “are” revised to “were”.

Line 322, “in the upstream” revised to “upstream”.

Line 323, “are” revised to “were”.

Line 323, Add “the”.

Line 323, “Transposase” revised to “transposase”.

Line 324, “share” revised to “shared”.

Line 324, “It is worth noting that” revised to “Notably,”.

Line 324, “is” revised to “was”.

Line 327, “transposase” revised to “transposases,”.

Line 327, Add “the”.

Line 328, “occurred” revised to “underwent”.

Line 329, Add “the”.

Line 330, “mediates” revised to “exhibits”.

Line 331, “fluoroquinolones” revised to “fluoroquinolone”.

Line 331, “an” revised to “a”.

Line 331, “ β - Lactamase” revised to “ β -lactamase”.

Line 332, “*Tet(A)*” revised to “*tet(A)*”.

Line 332, Add “a”.

Line 332, “Gram” revised to “gram”.

Line 333, Add “the”.

Line 333, “by” revised to “via”.

Line 335, “be” revised to “represent”.

Line 336, “incompatible” revised to “incompatibility”.

Line 336, Add “the”.

Line 337, “popular” revised to “distributed”.

Line 339, Add “in”.

Line 340, “Plasmid” revised to “The plasmid”.

Line 340, Add “the”.

Line 341, “gene *repA*” revised to “the *repA* gene”.

Line 341, “multi-drug” revised to “multidrug”.

Line 341, “resistant” revised to “resistance”.

Line 342, Add “the”.

Line 343, Add “have undergone”.

Line 345, Add “the”.

Line 345, “is” revised to “was”.

Line 346-347, “which mediates the resistance of macrolides” revised to “which mediates resistance to macrolides”.

Line 348, “exists an” revised to “an”.

Line 349, “the” revised to “a”.

Line 349, Add “exists”.

Line 349, “is” revised to “has been”.

Line 350, “classical” revised to “classic”.

Line 351, “,” revised to “;”.

Line 351, “Its” revised to “, its”.

Line 351, Add “regions”.

Line 351, Add “The”.

Line 352, “locates in” revised to “are located”.

Line 352, “,” revised to “;”.

Line 358, “Island” revised to “island”.

Line 359, Add “events”.

Line 359, “It is worth noting that” revised to “Notably,”.

Line 362, “is” revised to “are”.

Line 362, “at its upstream” revised to “, upstream”.

Line 363, Add “of this gene”.

Line 364, “them the” revised to “which”.

Line 365, “The same as” revised to “Similar to”.

Line 365, “carries” revised to “carried”.

Line 366-367, “the phage-carrying” revised to “phage-carrying”.

Line 367, “Plasmid” revised to “The plasmid”.

Line 367, “IncR,” revised to “the IncR type,”.

Line 368, “For the clinical” revised to “Among clinical”.

Line 369, “drug-resistant genes of” revised to “drug resistance gene”.

Line 371, “-resistant genes” revised to “resistance genes”.

Line 372, “-resistant genes” revised to “resistance genes”.

Line 374, “mainly distributed” revised to “distributed mainly”.

Line 375, “some” revised to “to a lesser extent”.

Line 375, Add “plasmid”.

Line 376, “plasmid with” revised to “to”.

Line 376, “has been” revised to “was”.

Line 376, “,” revised to “;”.

Line 378, “the homologous plasmids with” revised to “plasmids homologous to”.

Line 379, “and were isolated” revised to “isolated”.

Line 380, “it was reported” revised to “reported”.

Line 384, “overlap” revised to “overlapped”.

Line 385, “-resistant” revised to “resistance”.

Line 386, Add “the”.

Line 386, “-resistant” revised to “resistance”.

Line 386, “are” revised to “were”.

Line 386, “recent” revised to “the last”.

Line 388, “ESBLs” revised to “ESBL-carrying”.

Line 389, Add “the”.

Line 389, “ESBLs” revised to “ESBL”.

Line 390, “Plasmids” revised to “The plasmids”.

Line 390-391, “ESBLs” revised to “ESBL-carrying”.

Line 392, “are” revised to “were”.

Line 392, Add “strains,”.

Line 393, “plasmid has” revised to “plasmids shared”.

Line 395, “resistant” revised to “resistance”.

Line 395, “multi-drug resistant” revised to “multidrug resistance”.

Line 397, “multi-drug” revised to “multidrug”.

Line 399, “multi-drug” revised to “multidrug”.

Line 400, “-resistant genes” revised to “resistance genes”.

Line 401, “AMRs” revised to “AMR”.

Line 401, “between” revised to “among”.

Line 411, “bacteria” revised to “bacterial”.

Line 412, Add “the”.

Line 414, “Competing interests:” revised to “Competing interests:”.

Line 415, “Authors declare that they have no competing interests.” revised to “The authors declare that they have no competing interests.”.

Line 417-419, “All accession numbers to bacteria genome or plasmids relating to the paper and deposited in a GenBank database. All data are available in the main text or supplementary materials.” revised to “All accession numbers for bacterial genomes or plasmids related to the paper were deposited in a GenBank database (Table S1 and Table S2). All the data are available in the main text or supplementary materials.”.

Line 670, “multi-drug resistant” revised to “the multidrug resistance”.

Line 690, Add “the”.

Line 691, Add “the”.

Line 691, Add “the”.

Line 692, “red” revised to “Red”.

Line 712, Add “the”.

Line 713, “schematic” revised to “Schematic”.

Line 713, “comparison” revised to “Comparison”.

Line 713, Add “the”.

Line 714, “schematic” revised to “Schematic”.

Line 714, Add “the”.

Line 715, “red” revised to “Red”.

Line 725, “plasmids” revised to “the plasmid”.

Line 725, Add “and”.

Line 726, Add “the”.

Line 727, Add “the”.

Line 730, Add “and”.

Line 730-731, Add “in the”.

Line 751, “phenotype” revised to “phenotypes”.

Line 751, “Non-susceptible phenotype” revised to “Nonsusceptible phenotype”.

Line 751, “Chuckar” revised to “Chukar”.

Line 751, “ST Type” revised to “ST”.

February 4, 2023

Prof. Chengmin Wang
Guangdong Academy of Sciences
Institute of Zoology, Guangdong Key Laboratory of Animal Conservation and Resource Utilization
Guangzhou, Guangdong
China

Re: Spectrum02691-22R2 (Genomic characteristics and molecular epidemiology of multidrug resistant *Klebsiella pneumoniae* carried by wild birds)

Dear Prof. Chengmin Wang:

Your manuscript has been accepted, and I am forwarding it to the ASM Journals Department for publication. You will be notified when your proofs are ready to be viewed.

Sincerely,

Diyan Li
Editor, Microbiology Spectrum
